# Quantitative mapping of proteasome interactomes and substrates using ProteasomeID

**Aleksandar Bartolome, Julia C Heiby[†], Domenico Di Fraia[†], Ivonne Heinze, Hannah Knaudt, Ellen Spaeth, Omid Omrani, Alberto Minetti, Maleen Hofmann, Joanna M Kirkpatrick, Therese Dau\*, Alessandro Ori\***

Leibniz Institute on Aging - Fritz Lipmann Institute, Jena, Germany

**Abstract** Proteasomes are essential molecular machines responsible for the degradation of proteins in eukaryotic cells. Altered proteasome activity has been linked to neurodegeneration, auto-immune disorders and cancer. Despite the relevance for human disease and drug development, no method currently exists to monitor proteasome composition and interactions in vivo in animal models. To fill this gap, we developed a strategy based on tagging of proteasomes with promiscuous biotin ligases and generated a new mouse model enabling the quantification of proteasome interactions by mass spectrometry. We show that biotin ligases can be incorporated in fully assembled proteasomes without negative impact on their activity. We demonstrate the utility of our method by identifying novel proteasome-interacting proteins, charting interactomes across mouse organs, and showing that proximity-labeling enables the identification of both endogenous and small-molecule-induced proteasome substrates.

**\*For correspondence:**
therese.dau@leibniz-fli.de (TD);
alessandro.ori@leibniz-fli.de (AO)

[†]These authors contributed equally to this work

## eLife assessment

This study presents an **important** method and resource in cell lines and in mice for mass spectrometry-based identification of interactors of the proteasome, a multi-protein complex with a central role in protein turnover in almost all tissues and cell types. The method presented, including the experimental workflow and analysis pipeline, as well as the several lines of validation provided throughout, is **convincing**. Given the growing interest in protein aggregation and targeted protein degradation modalities, this work will be of interest to a broad spectrum of basic cell biologists and translational researchers.

## Introduction

The ubiquitin–proteasome system (UPS) is a major selective protein degradation system of eukaryotic cells (*Schmidt and Finley, 2014*). Proteasomes influence crucial processes in the cell including protein homeostasis, DNA repair, signal transduction and immune responses (*Rousseau and Bertolotti, 2018*) by degrading a multitude of regulatory, short-lived, damaged and misfolded proteins (*Glickman and Ciechanover, 2002*; *Abildgaard et al., 2020*). Impairment of proteasome function has been described to occur during aging *Saez and Vilchez, 2014* and in neurodegenerative diseases *Tanaka and Matsuda, 2014*; *Schmidt and Finley, 2014*, and high expression levels of proteasome subunits have been linked to longevity in different species including worms (*Vilchez et al., 2012*), fish (*Kelmer Sacramento et al., 2020*), and flies (*Tonoki et al., 2009*; *Nguyen et al., 2019*). While partial proteasome inhibition is sufficient to induce cellular senescence (*Chondrogianni et al., 2003*; *Chondrogianni and Gonos, 2004*), prolonged inhibition of the proteasome induces apoptosis (*Abbas and*

*Larisch, 2021*). This has been exploited in cancer therapies, for example against blood cancers, as cancer cells are especially sensitive to proteotoxic stress (*Manasanch and Orlowski, 2017*). Furthermore, the role of proteasomes extends beyond the degradation of proteins. In the immune system, peptide generation by specialized immunoproteasomes regulates antigen presentation and has been linked to auto-immune disorders (*Feist et al., 2016*).

The eukaryotic proteasome consists of two subcomplexes: the 'core particle' (20 S proteasome) that can be present either alone or in association with one (26 S proteasome) or two 19 S 'regulatory particles' (30 S proteasome). These proteasome variants are considered constitutive and they are generated in most of the cells. In addition to the 19S regulatory particles, the core particle can be associated with different alternative regulatory complexes including the proteasome activator PA200 (named PSME4 in mammals) or three different versions of the 11S regulator complex PA28 – PA28α (PSME1), PA28β (PSME2), and PA28γ (PSME3). These are ATP-independent proteasome regulators that modulate proteasome activity towards small peptides and unfolded proteins and can facilitate protein degradation independent of ubiquitin (*Toste Rêgo and da Fonseca, 2019*; *Mao et al., 2008*).

The composition of cellular proteasomes can be heterogeneous due to the assembly of distinct sub-complexes in variable stoichiometries, and to the incorporation of tissue/cell type specific components. Specialized proteasomes have been described in immune cells, thymus, sperm cells (*Abi Habib et al., 2022*), and cancer (*Javitt et al., 2023*). Sub-populations of proteasomes have been shown to be located in different cellular compartments including inner nuclear membrane (*Albert et al., 2017*), endoplasmic reticulum (*Albert et al., 2020*), Golgi apparatus (*Eisenberg-Lerner et al., 2020*), plasma membranes (*Ramachandran and Margolis, 2017*), primary cilia *Gerhardt et al., 2016*, and protein aggregates (*Guo et al., 2018*). Moreover, the assembly state of proteasomes and their biophysical properties can be modulated in response to, for example hyperosmotic stress (*Yasuda et al., 2020*) or amino acid starvation (*Uriarte et al., 2021*).

Due to the dynamic nature of proteasome composition and its distribution within cells, it is important to employ analytical approaches that can capture protein-protein interactions across different cellular compartments and account for transient interactions. Traditional methods such as co-immunoprecipitation, affinity purification or cell fractionation coupled to mass spectrometry have been valuable in studying proteasome interactions (*Fabre et al., 2013*; *Fabre et al., 2015*; *Bousquet-Dubouch et al., 2009*; *Geladaki et al., 2019*; *Wang et al., 2007*), but they may not capture transient or weak interactions. To address this limitation, proximity labeling assays, such as BioID or APEX, utilize engineered proteins that can biotinylate neighboring proteins in close proximity within the cellular context, and thereby enable a more comprehensive and dynamic characterization of protein-protein interactions (*Gingras et al., 2019*).

Here, we present an approach named ProteasomeID that enables the quantification of proteasome interacting proteins and substrates in cultured human cells and mouse models. The approach entails fusing the proteasome with promiscuous biotin ligases, which are incorporated into fully assembled proteasomes without affecting their activity. The ligases label proteins with biotin that come into proximity (~10 nm) of the tagged proteasome subunit. Biotinylated proteins are then captured from cell or tissue lysates using an optimized streptavidin enrichment protocol and analyzed by deep Data Independent Acquisition (DIA) mass spectrometry. We show that ProteasomeID can quantitatively monitor the majority of proteasome interacting proteins both in situ and in vivo in mice, and identify novel proteasome interacting proteins. In addition, when combined with proteasome inhibition, ProteasomeID enables the identification of proteasome substrates, including low abundant transcription factors.

## Results

### Design of a proximity labeling strategy to monitor proteasome interactions

First, we tested multiple locations of the biotin ligase to ensure that the integration within the proteasome complex would not impede its assembly or disrupt its functionality. Based on previous studies where proteasome members were tagged with fluorescent proteins in mammalian cells (*Salomons et al., 2010*; *Bingol and Schuman, 2006*), we fused the promiscuous biotin ligase BirA* either to the C-termini of the 20 S core particle protein PSMA4/α3, the 19 S particle base protein PSMC2/

Rpt1 or the N-terminus of the 19 S particle lid protein PSMD3/Rpn3 (*Figure 1a* and *Figure 1—figure supplement 1a*). Each construct also contained a FLAG tag for fusion protein detection. We used these constructs to generate stable HEK293 FlpIn TREx (HEK293T) cell lines that overexpress the BirA* fusion proteins under the control of a tetracycline inducible promoter. A cell line expressing only the BirA* protein was used as a control to account for non specific biotinylation. We confirmed tetracycline-dependent expression of the corresponding cell lines by anti-FLAG immunoblot, and confirmed biotinylating activity following supplementation of exogenous biotin using streptavidin-HRP blot (*Figure 1b*). These results were validated by immunofluorescence analysis (*Figure 1c* and *Figure 1—figure supplement 1b*).

In order to identify the most suitable fusion protein for proximity labeling of proteasomes, we compared the enrichment of proteasome components in streptavidin pull downs performed with different cell lines that we generated. We optimized a BioID protocol that we previously developed (*Mackmull et al., 2017*) to improve the identification of biotinylated proteins by liquid chromatography tandem mass spectrometry (LC-MS/MS). Briefly, the protocol entails the capture of biotinylated proteins from cell lysates using streptavidin beads followed by enzymatic on-bead digestion and analysis of the resulting peptides by LC-MS/MS. We introduced chemical modification of streptavidin beads and changed the protease digestion strategy to reduce streptavidin contamination following on-bead digestion (*Figure 1—figure supplement 2a*). In addition, we improved the data analysis by implementing Data Independent Acquisition (DIA). These optimizations allowed us to drastically reduce (>fourfold) the background from streptavidin-derived peptides (*Figure 1—figure supplement 2b and c*), and to increase more than twofold the number of identified proteins and biotinylated peptides in our BioID experiments (*Figure 1—figure supplement 2d*).

Using this optimized BioID protocol, we analyzed samples enriched from cell lines expressing PSMA4-BirA*, PSMC2-BirA*, BirA*-PSMD3 or BirA* control (*Figure 1—figure supplement 2e*). We found significant enrichment of proteasome subunits for each of the cell lines expressing BirA* fusion proteins compared to BirA* control (*Figure 1—figure supplement 2f* and *Supplementary file 1*). However, the pattern of enrichment varied between fusion proteins (*Figure 1d and e*). PSMA4-BirA* provided the strongest enrichment for 20 S proteins, but also a prominent enrichment of other proteasome components (typically >4 fold), while PSMC2-BirA* enriched preferentially 19 S base proteins. BirA*-PSMD3 displayed a more homogenous, but less pronounced enrichment of proteasome proteins (typically ~twofold). The different enrichment patterns reflect the localization of the fusion proteins within the complex, but it might also indicate interference of the biotin ligase with the assembly of the proteasome, especially in the case of PSMC2-BirA*. Consequently, we decided to focus on the PSMA4-BirA* fusion protein for further characterization as it showed an overall stronger enrichment of proteasomal proteins from both the 20 S and 19 S particles.

## PSMA4-BirA* is incorporated into fully assembled proteasomes and does not interfere with their proteolytic activity

Next, we performed a series of experiments to confirm the incorporation of PSMA4-BirA* into fully assembled proteasomes and exclude any potential interference with their proteolytic activity. Since the BirA* fusion protein is over-expressed using a CMV promoter, we first confirmed that the abundance levels of PSMA4-BirA* are comparable to the ones of the endogenous PSMA4 using an anti-PSMA4 immunoblot (*Figure 2a* and *Figure 2—figure supplement 1a*).

In order to assess the assembly of PSMA4-BirA* into proteasome complexes, we performed size exclusion chromatography coupled to quantitative mass spectrometry (SEC-MS) analysis of the cell line expressing PSMA4-BirA* following induction by tetracycline. Protein elution profiles built from mass spectrometry data obtained from 32 SEC fractions revealed three major distinct peaks corresponding to the major assembly states of the proteasome (*Supplementary file 2*). These include 30 S proteasomes, containing a core particle capped with two regulatory particles, 26 S proteasomes, containing a core particle capped with 1 regulatory particle, and isolated core particles (20 S proteasomes; *Figure 2b* and *Figure 2—figure supplement 1b*). With the exception of a peak in lower molecular weight fractions (~fraction 20), likely representing intermediate complex assemblies, the majority of the BirA* signal (~60%) correlated with the elution profile of other proteasome components in all the assembly states (*Figure 2—figure supplement 1b*). We have additionally performed anti-FLAG immunoblot upon blue native gel electrophoresis (BN-PAGE) to confirm the incorporation

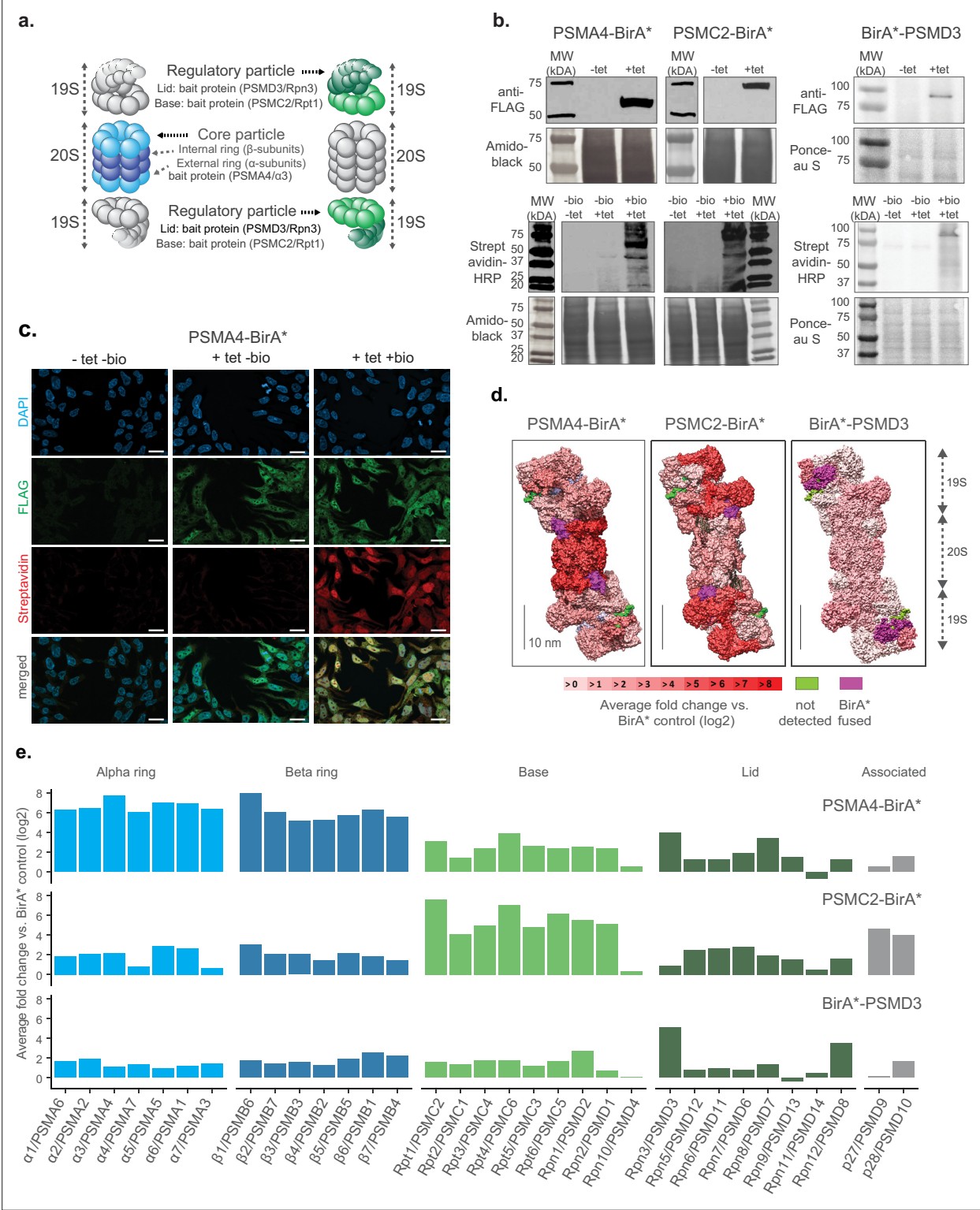

**Figure 1.** Establishment of a cell culture model system for proximity labeling of proteasomes. (**a**) Schematic representation of proteasome with the substructures containing subunits fused to biotin ligase highlighted in color (shown in 2 shades of blue and green for 20 S and 19 S proteasome, respectively). (**b**) Upper panel, immunoblot of BirA* fusion proteins performed on lysates from HEK293T cells stably transfected with PSMA4-BirA*-FLAG, PSMC2-BirA*-FLAG or BirA*-FLAG-PSMD3 following 24 hr incubation with (+tet) or without (−tet) tetracycline. Lower panel, streptavidin-HRP blot following induction of BirA* fusion proteins with tetracycline and supplementation of biotin for 24 hr. Amido Black or Ponceau stainings were used as loading control. HRP: horseradish peroxidase. (**c**) Immunofluorescence analysis of PSMA4-BirA*-FLAG cell line 4 days after seeding without addition of

*Figure 1 continued on next page*

*Figure 1 continued*

any substance (-tet -bio), with addition of only tetracycline for 4 days (+tet bio) or with addition of both tetracycline for 4 days and biotin for 1 day (+tet + bio). Scale bar = 20 μm. (**d**) Level of enrichment of proteasome components measured by ProteasomeID in the context of the proteasome structure. Enriched proteins are depicted in different shades of red according to the log2 fold enrichment vs. BirA* control. Scale bar = 10 nm. The proteasome structure depicted was obtained from the PDB:5T0C model of the human 26 S proteasome *Chen et al., 2016* and rendered using Chimera (*Pettersen et al., 2004*). (**e**) Enrichment level comparison for proteasome components achieved in 3 different cell lines of ProteasomeID. Enriched proteins are depicted in the same color code as in panel a. and according to the log2 fold enrichment vs. BirA* control. n=4 biological replicates.

The online version of this article includes the following source data and figure supplement(s) for figure 1:

**Source data 1.** Raw unedited gels for *Figure 1*.

**Source data 2.** Uncropped and labeled gels for *Figure 1*.

**Figure supplement 1.** Further characterization of PSMC2-BirA*,BirA*-PSMD3 expressing cell lines.

**Figure supplement 2.** Optimization of BioID workflow and ProteasomeID cell lines evaluation.

**Figure supplement 3.** Plasmid map for construct PSMC2-BirA*.

of PSMA4-BirA* into assembled proteasomes, and to demonstrate that the assembly of proteasomes is not affected by the overexpression of PSMA4-BirA* (*Figure 2c*). Finally, we used our mass spectrometry data to identify sites of protein biotinylation in PSMA4-BirA* expressing cells. By mapping the identified sites on the proteasome structure, we could confirm the specificity of protein biotinylation by showing that 25 out of 26 residues identified as biotinylated by PSMA4-BirA* are located less than 10 nm away from the C-terminus of PSMA4 (*Figure 2d* and *Supplementary file 2*). Importantly, 18 of these sites were located on proteins from the 19 S regulatory particle, further supporting the incorporation of PSMA4-BirA* into assembled proteasomes.

In order to test the influence of BirA* fusion proteins on proteasome function, we measured proteasome chymotrypsin-like activity in cell lysates from BirA* expressing cell lines in presence or absence of tetracycline. We observed a slight, not-significant reduction of proteasome activity (~15–20%) following addition of tetracycline that was comparable between cell lines expressing proteasome PSMA4-BirA* or BirA* control (*Figure 2e*). In addition, we tested the impact of PSMA4-BirA* overexpression on the degradation of c-Myc, a known proteasome substrate, using a cycloheximide chase experiment. We showed that tetracycline-mediated induction of PSMA4-BirA* did not have a major effect on the degradation kinetics of c-Myc (*Figure 2f* and *Figure 2—figure supplement 1c*). Also, levels of K48-ubiquitylated proteins were not affected (*Figure 2—figure supplement 1d*), further confirming that PSMA4-BirA* has no major impact on the proteolytic activity of the proteasome in cells.

## ProteasomeID retrieves proteasome subunits, assembly factors and known proteasome interactors

To assess the ability of ProteasomeID to retrieve known proteasome interacting proteins (PIPs) in addition to proteasome components, we implemented a logistic regression classifier algorithm (*Figure 3a*). The classifier is based on an 'Enrichment score' that we obtained by combining the average log2 ratio and the negative logarithm of the q value from differential protein abundance analysis performed vs. BirA* control line. We then used a set of true positives, here, proteasome members, and true negatives, here, mitochondrial matrix proteins, which should not interact directly with the proteasome under homeostatic conditions, to identify a cut-off for defining ProteasomeID-enriched proteins at a controlled false positive rate (*Figure 3b*). We validated the classifier using a receiver operator characteristic (ROC) analysis (*Figure 2—figure supplement 1e*). Using the classifier, we identified 608 protein groups enriched by ProteasomeID for the PSMA4-BirA* dataset (false positive rate, FPR <0.05, *Figure 3c* and *Supplementary file 3*).

We investigated the occurrence of PIPs among ProteasomeID-enriched proteins using the two approaches. First, we compared the BioID data from PSMA4-BirA* to the SEC-MS data obtained from the same cell line. We could show that SEC elution profiles of proteins enriched in ProteasomeID tend to have significantly higher correlation with the elution profile of PSMA4 (*Figure 3d*). Since correlated SEC elution profiles are typically interpreted as an indication of physical association in this type of experiment, these data suggest that ProteasomeID can enrich PIPs. Second, we compared our dataset to these three previous proteomic studies that investigate PIPs using complementary

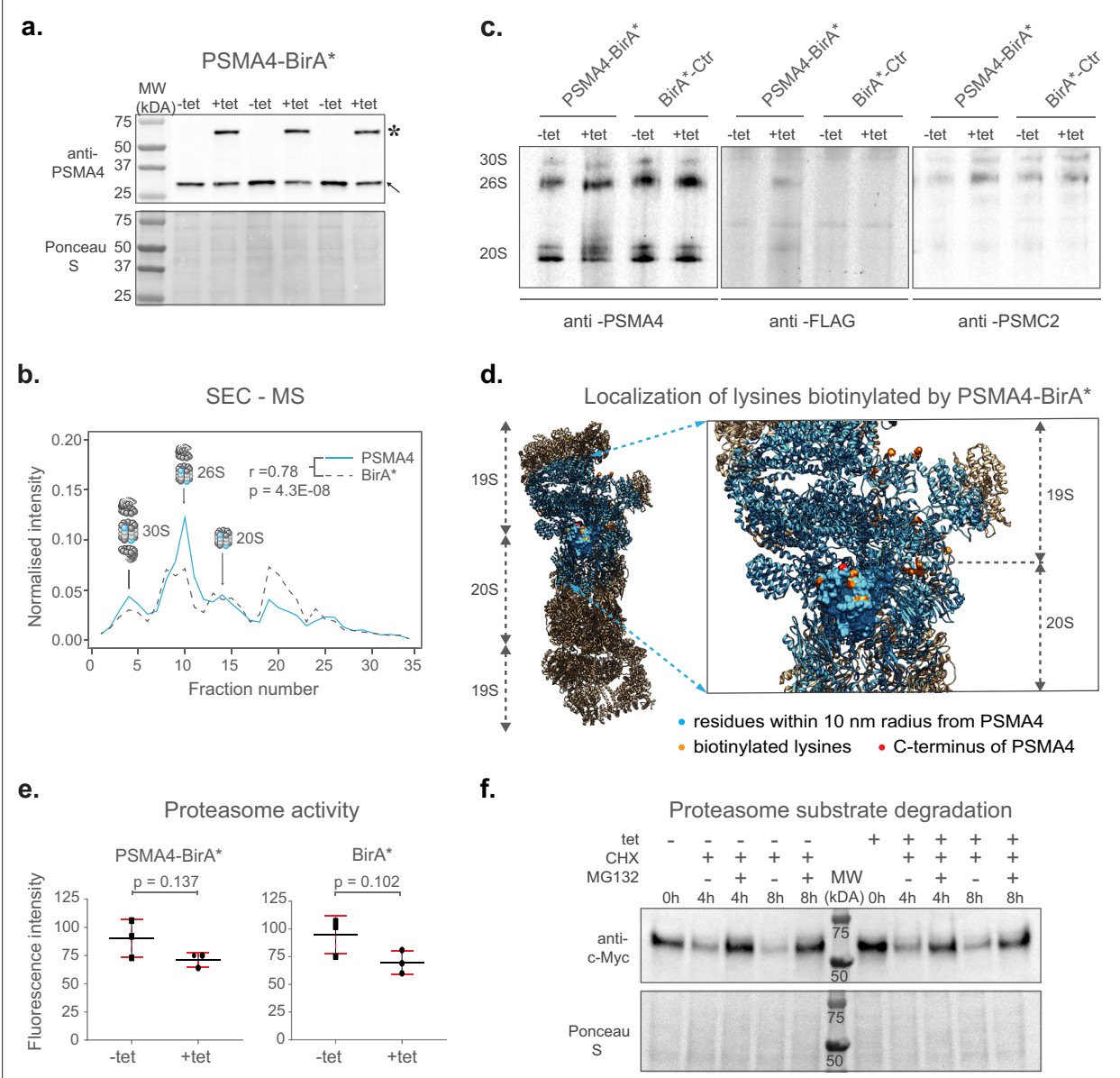

**Figure 2.** Validation of ProteasomeID. (**a**) Comparison of expression levels of PSMA4-BirA* (lanes marked by star) and its endogenous counterpart (lanes marked by arrowhead), following 24 hrincubation with (+tet) or without (−tet) tetracycline. Ponceau S staining was used as loading control. (**b**) Size exclusion chromatography (SEC) analysis of lysates from HEK293T cells stably expressing PSMA4-BirA* following 24 hr incubation with tetracycline. SEC fractions were analyzed by DIA mass spectrometry and elution profiles were built for each protein using protein quantity values normalized to the sum of quantities across all fractions. Depicted are elution profiles of PSMA4 (proteasome subunit, blue) and BirA* (biotinylating enzyme, dashed line). The peaks corresponding to different proteasome assemblies were assigned based on the elution profiles of other proteasome components. (**c**) Immunoblot for proteasome subunits PSMA4 (left panel), PSMC2 (right panel), and FLAG tag (middle panel) of cell lysates separated by native PAGE from PSMA4-BirA* and BirA*-Ctr cell lines with and without tetracycline addition. Tet = tetracycline, 30S=indicates position of proteasome structures containing one core and 2 regulatory particles, 26S=indicates position of proteasome structures containing one core and 1 regulatory particle, 20S=indicates position of proteasome structures consisting of only single core particle. (**d**) Biotinylated lysines identified by ProteasomeID. All the residues within a 10 nm radius of the PSMA4 C-terminus are highlighted in cyan. Red color indicates the C-terminus of PSMA4 where BirA* is fused (not present in the structure), and the identified biotinylated lysines are depicted in orange. Only the structure of the modified subunit is depicted with a surface model and all the other subunits are depicted as helix-loop structures. Biotinylated residues were obtained from the ACN fraction of PSMA4-BirA*. The proteasome structure depicted was obtained from the PDB:5T0C model of the human 26 S proteasome (*Chen et al., 2016*) and rendered using Chimera (*Pettersen et al., 2004*). (**e**) Proteasome activity assay performed on lysates from cell lines expressing different BirA* fusion proteins, following 24 hr incubation with (+tet) or without (−tet) tetracycline. Equal amounts of protein extracts were incubated with proteasome substrate LLVY-7-Amino-4-methylcoumarin (AMC) and substrate cleavage assessed by fluorimetry. n=3 biological replicates, error bars indicate standard deviation of the mean, paired t-test.

*Figure 2 continued on next page*

*Figure 2 continued*

(**f**) Cycloheximide-chase experiment on c-Myc stability. PSMA4-BirA*cells were incubated with 50 µg/ml cycloheximide (CHX) for the indicated times in the presence or absence of MG132 (20 µM) and tetracycline (1 µg/µl). Cells lysates were then prepared for western blot analysis of steady-state levels of c-Myc.Tet=tetracycline, CHX = cycloheximide.

The online version of this article includes the following source data and figure supplement(s) for figure 2:

**Source data 1.** Raw unedited gels for *Figure 2*.

**Source data 2.** Uncropped and labeled gels for *Figure 2*.

**Figure supplement 1.** ProteasomeID cell line validation.

**Figure supplement 1—source data 1.** Raw unedited gels for *Figure 2—figure supplement 1*.

**Figure supplement 1—source data 2.** Uncropped and labeled gels for *Figure 2—figure supplement 1*.

approaches: *Fabre et al., 2015* used fractionation by glycerol gradient ultracentrifugation, *Bousquet-Dubouch et al., 2009* used immunoaffinity purification, and (*Geladaki et al., 2019*) applied cellular fractionation by differential centrifugation. We found 72 proteins overlapping to our dataset including almost all known proteasome members, assembly factors, and activators (*Figure 3e and f*). The overlapping proteins also include members of other complexes known to interact with the proteasome, that is the SKP1-CUL1-F-box protein (SCF) complex (CUL1 and FBOX7), and the chaperonin containing TCP-1 (CCT) complex (CCT2, CCT3, CCT5 and CCT7). Some proteins common to at least two of previous studies but were missing in our dataset include immunoproteasome subunits (PSMB10/LMP10, PSMB8/LMP7) that are not expressed in HEK293 cells, the shuttling factor RAD23B and ubiquitin-related proteins including USP14, UCHL5 and UBE3C. Manual inspection of proteins identified exclusively by ProteasomeID revealed additional 37 proteins that were reported in additional studies that investigated proteasome interactions (summarized in *Supplementary file 3*), and 19 proteins that have been associated to the ubiquitin proteasome system (UPS; *Figure 2—figure supplement 1f*). In addition, another 176 proteins found in the ProteasomeID network are interactors of previously reported proteasome interacting proteins, according to high confidence interactions from the STRING database (*Snel et al., 2000*; *Figure 3g*). Therefore, in total, 304 out of 608 (50%) ProteasomeID enriched proteins could be matched to known proteasome-interacting proteins or their direct binding partners. The remaining 304 proteins, which are enriched in GO terms related to microtubule and cilia organization (*Supplementary file 3*), likely represent novel interacting proteins, or false positives due to the presence of a subpopulation of unassembled PSMA4-BirA*. Comparing the enrichment scores, however, suggest that they are likely proteins that come into proximity of the proteasome without physically interacting with it (*Figure 2—figure supplement 1g*).

## Identification of proteasome substrates by ProteasomeID

Having demonstrated that ProteasomeID can be used to obtain snapshots of the proteasome-proximal proteome, we next investigated whether we could use this approach to identify proteasome substrates. We reasoned that under steady state conditions the interaction between proteasomes and their substrates might be too short lived to enable efficient biotinylation. Furthermore, the proteolytic cleavage of substrates by the proteasome would eventually make biotinylated peptides impossible to be identified by our standard proteomic workflow based on tryptic peptides. Therefore, we generated HEK293T cell lines expressing PSMA4-miniTurbo or miniTurbo alone, enabling shorter biotinylation time thanks to the enhanced activity of miniTurbo as compared to BirA* (*Branon et al., 2018*; *Figure 4—figure supplement 1a and b*). We tested biotinylation by PSMA4-miniTurbo using immunoblot and confirmed that 2 hr were sufficient to achieve biotinylation levels comparable to the one PSMA4-BirA* after 24 hr of biotin supplementation (*Figure 4—figure supplement 1c*). Furthermore, we confirmed enrichment of proteasome members and interacting proteins (*Figure 4—figure supplement 1d*), and observed positive correlation between the enrichments measured using PSMA4-miniTurbo and PSMA4-BirA*, relatively to their respective control lines (*Figure 4—figure supplement 1e*). Applying the classifier algorithm, we identified 168 protein groups enriched for PSMA4-miniTurbo (FPR <0.05, *Figure 4—figure supplement 1f* and g).

Next, we included a step of acute inhibition of the proteasome by the potent cell-permeable inhibitor MG132 for 4 hr (*Figure 4a*). We confirmed proteasome inhibition by the accumulation of ubiquitylated proteins assessed by immunoblot (*Figure 4—figure supplement 1h*). Principal component

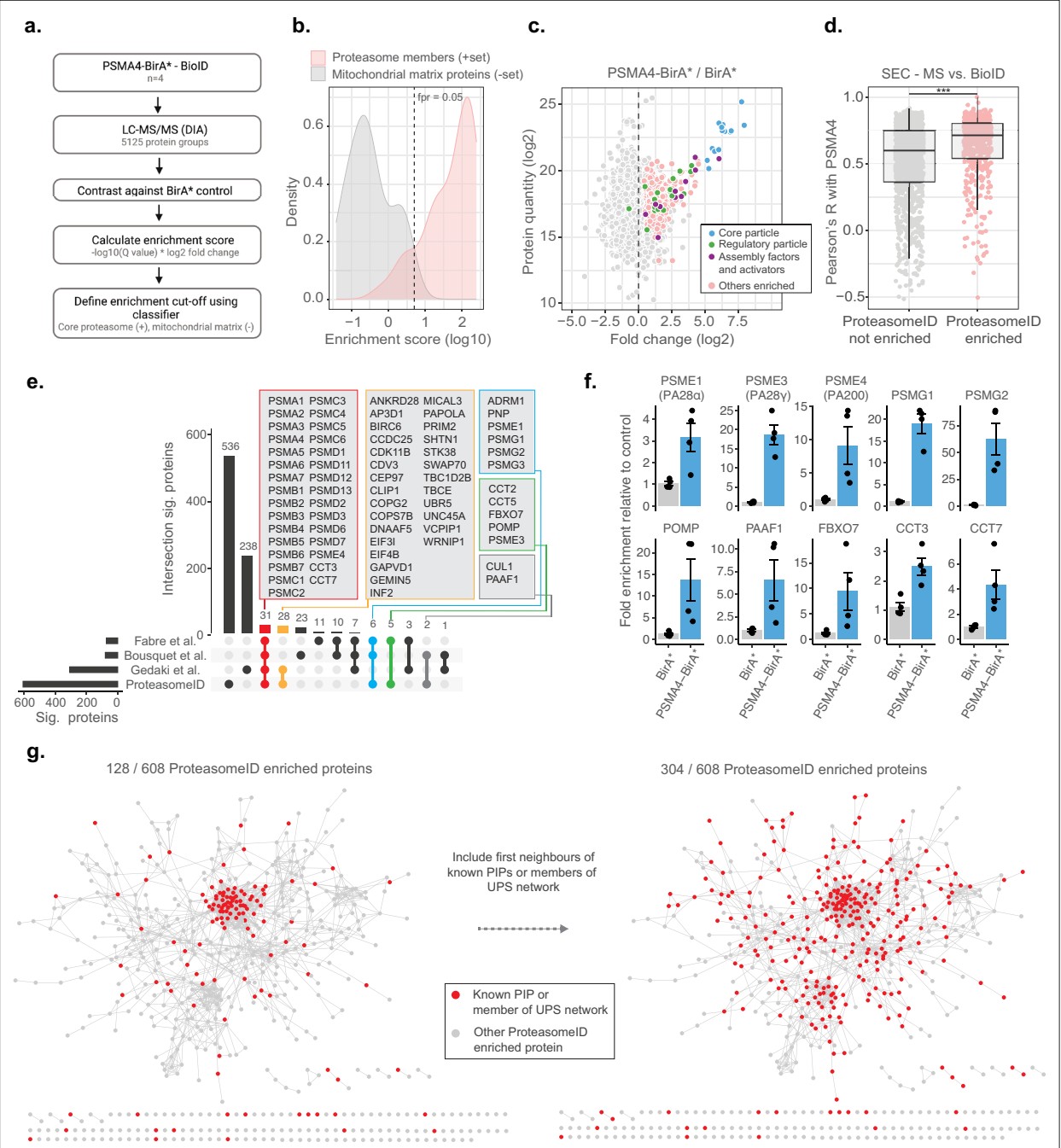

**Figure 3.** ProteasomeID identifies known proteasome interactors. (**a**) Schematic depiction of the classifier algorithm used to unbiasedly define proteins enriched by ProteasomeID. The classifier is based on an 'Enrichment score' obtained by combining the average log2 ratio and the negative logarithm of the q value from differential protein abundance analysis performed vs. BirA* control line. (**b**) Distribution of enrichment scores calculated by the classifier algorithm for proteasome subunits (set of true positives) and mitochondrial matrix proteins (set of true negatives). The dashed vertical line indicates the enrichment score cut-off to define ProteasomeID enriched proteins at FPR <0.05. (**c**) MA plot of proteins enriched by streptavidin pull-down and analyzed by DIA mass spectrometry. Highlighted in color are proteasome members, assembly factors and activators, and other proteins significantly enriched in ProteasomeID (FPR <0.05). n=4 biological replicates.(**d**) Comparison of co-elution profiles obtained by SEC-MS and proteins enriched in ProteasomeID. Pearson correlation values were calculated between PSMA4 and all the other proteins quantified in SEC-MS (n=4680). Correlation values were compared between proteins significantly enriched in PSMA4-BirA* vs. BirA* and all the other proteins quantified in the ProteasomeID experiment. *** p<0.001 Wilcoxon Rank Sum test with continuity correction. (**e**) Upset plot showing overlap between ProteasomeID enriched proteins and previous studies that investigated proteasome interacting proteins (PIPs). Different subsets of overlapping proteins are highlighted in color framed boxes. (**f**) Bar plots comparing the levels of enrichment obtained in ProteasomeID experiment for proteasome activators, assembly factors and known

*Figure 3 continued on next page*

*Figure 3 continued*

PIPs. Enrichment levels were normalized to the levels detected in BirA* control cell line which was set to 1. Protein quantities were derived from DIA mass spectrometry data. Data are shown as mean ± standard error from n=4 biological replicates.(**g**) Network analysis of 608 interactors of PSMA4-BirA* obtained by ProteasomeID. Identified proteins were filtered for significance by a cutoff of log2 fold change >1 and Q value <0.05 in relation to BirA* control. Nodes representing identified proteins that are known PIPs or members of ubiquitin proteasome system (UPS) (left network) or identified protein and their first interacting neighbor are known PIPs or members of UPS (right network) are highlighted in red color. Edges represent high confidence (>0.7) protein-protein interactions derived from the STRING database (*Snel et al., 2000*).

analysis of the mass spectrometry data obtained from streptavidin enriched proteins revealed a clear separation between samples treated with MG132 or vehicle control (*Figure 4—figure supplement 1i*). Comparison of ProteasomeID enriched proteins from cells treated with proteasome inhibitor versus vehicle control revealed a subset of 141 proteins enriched exclusively upon MG132 treatment (*Figure 4b and c*). These include proteasome activators (PSME1/PA28α, PSME2/PA28β, PSME3 / PA28γ) and ubiquitin (*Figure 4d* and *Supplementary file 4*), consistent with the recruitment of proteasome activators and direct ubiquitylation of proteasome members following inhibition (*Rechsteiner and Hill, 2005*). Of the remaining 133 proteins, 77 (58%) have been shown in a previous study *Trulsson et al., 2022* to display increased ubiquitylation (as assessed by GlyGly enrichment and MS analysis) upon proteasome inhibition, and 39 are reported proteasome substrates such as MYC, ATF4, and PINK1 (*Figure 4c and d* and *Supplementary file 4*). In line with the observation that this subset of proteins is enriched for proteasome substrates, their levels were found to increase relatively to the rest of the proteome upon treatment with MG132 (*Figure 4e*). Notably, when examining the enrichment of these proteins using ProteasomeID, more pronounced effect sizes were observed (*Figure 4e*). This suggests that the increased enrichment of these proteins cannot be solely attributed to elevated cellular levels, indicating a specific targeting of these proteins to the proteasome upon MG132.

In order to validate our findings, we selected three proteins exclusively enriched in ProteasomeID upon MG132 that were not previously reported as proteasome substrates (*Figure 4f*): the BRCA1-associated ATM activator 1 (BRAT1 also known as BAAT1) the armadillo containing protein 6 (ARMC6) and Tigger transposable element-derived protein 5 (TIGD5). While the function of ARMC6 is unknown, BRAT1 has been associated with neurodevelopmental and neurodegenerative disorders (*Srivastava et al., 2016*) and shown to play a role in DNA damage response (*So and Ouchi, 2011*) and RNA processing (*Cihlarova et al., 2022*). TIGD5 encodes protein with DNA-binding features. Although its exact function is not established, it has been suggested to operate as a tumor suppressor in ovarian cancer (*Dai et al., 2022*). We then performed a cycloheximide chase experiment and could validate them as proteasome substrates (*Figure 4g* and *Figure 4—figure supplement 1j*).

Finally, we investigated whether we could use ProteasomeID to identify selective induction of protein degradation by small molecules. For this purpose, we used KB02-JQ1, a well-characterized PROTAC that targets bromodomain-containing proteins (BRDs) for proteasomal degradation (*Zhang et al., 2019*). We pre-treated cells with KB02-JQ1 for 8 hr prior to proteasome inhibition and biotin supplementation (*Figure 5a*). Mass spectrometry analysis showed global changes induced by KB02-JQ1 either in presence or absence of MG132, as indicated by PCA (*Figure 4—figure supplement 1k*). Importantly, we could detect prominent enrichment of BRD containing proteins following treatment with KB02-JQ1 (*Figure 5b* and *Supplementary file 4*). The effect was more pronounced for BRD2 and BRD3 and less striking for BRD4 (*Figure 5c*), presumably reflecting different kinetics of induced degradation by KB02-JQ1 (*Zhang et al., 2019*). Notably, the enrichment of BRD2 and BRD3 was also detectable in absence of MG132, in contrast to endogenous substrates that become enriched only following proteasome inhibition (*Figure 5c*). Together, these data demonstrate that ProteasomeID can be used to detect both endogenous and protein degrader-induced substrates of the proteasome in cultured cells.

## A mouse model for in vivo ProteasomeID

Having established and validated proximity labeling of proteasomes in a cell culture model, we designed a strategy to implement ProteasomeID in a mouse model (*Figure 6a*). The mouse model was designed to express the 20 S proteasome core particle PSMA4 fused to the biotinylating enzyme miniTurbo and a FLAG tag for the detection of the fusion protein. We chose miniTurbo instead of BirA* because of its higher biotinylating efficiency (*Branon et al., 2018*). PSMA4-miniTurbo was inserted in

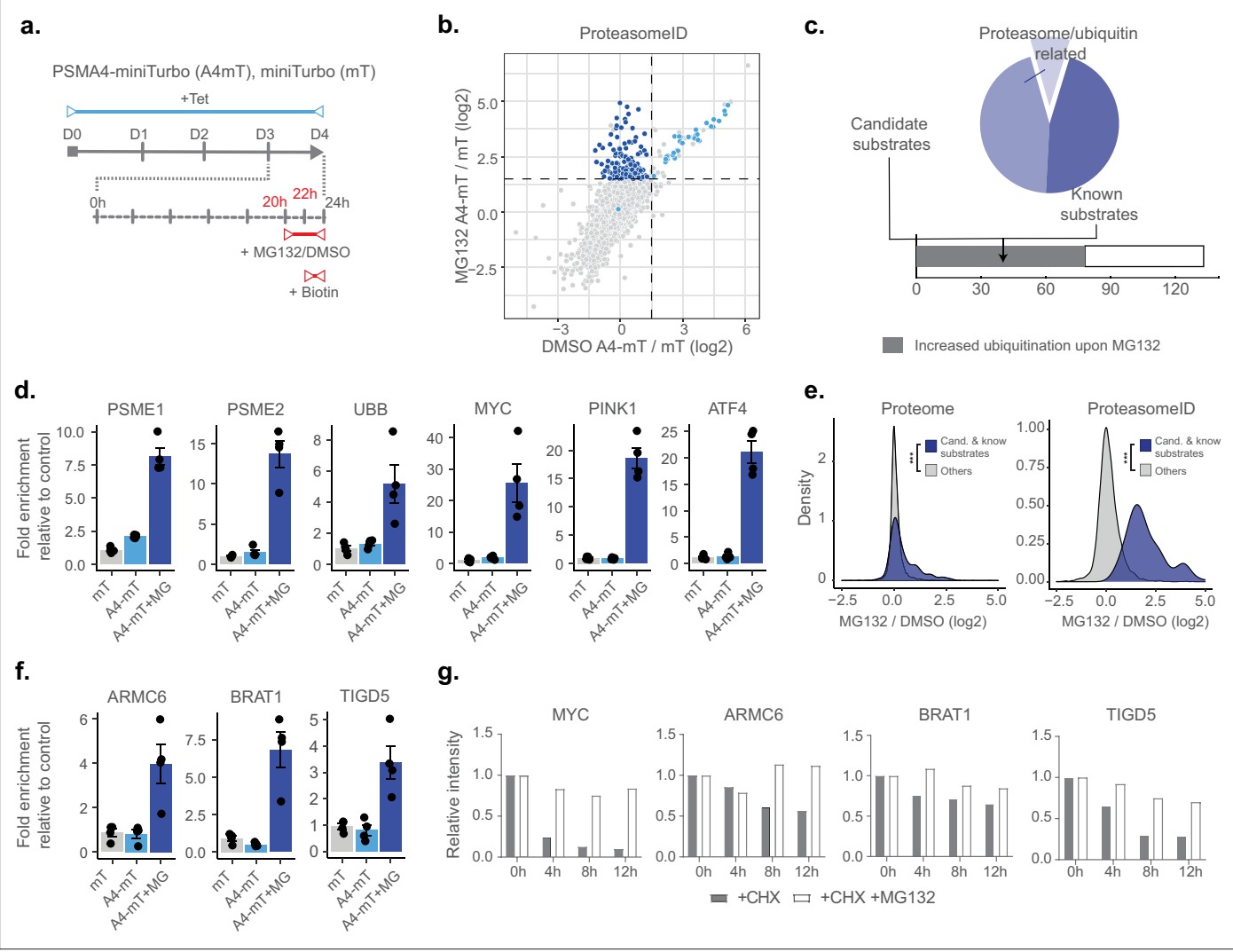

**Figure 4.** ProteasomeID identifies known and novel endogenous proteasome substrates. (**a**) Scheme of ProteasomeID workflow in HEK293T cells including proteasome inhibition by MG132. PSMA4-miniTurbo expression and incorporation into proteasomes is achieved by 4-day induction with tetracycline. Proteasome inhibition is achieved by addition of 20 μM MG132 4 hr before cell harvesting. Biotin substrate for miniTurbo is supplied 2 hr before cell harvesting. D: day; hr: hour; Tet: tetracycline; Bio: biotin. (**b**) Enriched proteins from ProteasomeID cells treated with proteasome inhibitor MG132 compared to vehicle control. The subset of proteins enriched exclusively upon MG132 treatment are highlighted in dark blue. Data were obtained from n=4 biological replicates. (**c**) Profile of proteins exclusively enriched upon MG132 treatment of ProteasomeID cells (pie chart). Identified proteins are represented by proteasome/ubiquitin related proteins, known proteasome substrates and potential previously unidentified substrates. The number of proteins identified by this approach for which previous studies showed increased ubiquitylation upon proteasome inhibition is shown in the lower bar plot. (**d**) Bar plots comparing the levels of proteasome activators and ubiquitin, and known proteasome substrates following streptavidin enrichment from different cell lines and following proteasome inhibition by MG132. Protein quantities were derived from DIA mass spectrometry data. mT: miniTurbo control cell line; A4-mT: PSMA4-miniTurbo cell line; I: proteasome inhibition by MG132. Data are shown as mean ± standard error from n=4 biological replicates. (**e**) Distribution of log2 fold changes following MG132 treatment for candidate and known proteasome substrates identified by ProteasomeID. The fold changes are compared to the other identified proteins using total proteome (left) or ProteasomeID data (right). *** p<0.001 Wilcoxon Rank Sum test with continuity correction. (**f**) Bar plots comparing the levels of three potential novel proteasome substrate proteins following streptavidin enrichment from different cell lines and following proteasome inhibition by MG132. Protein quantities were derived from DIA mass spectrometry data. mT: miniTurbo control cell line; A4-mT: PSMA4-miniTurbo cell line; I: proteasome inhibition by MG132. Data are shown as mean ± standard error from n=4 biological replicates. (**g**) Cycloheximide-chase experiment on stability of 3 potential novel proteasome substrate proteins. HEK293T cells were incubated with 50 μg/ml cycloheximide (CHX) for the indicated times in the presence or absence of MG132 (20 μM). Cell lysates were then prepared for western blot analysis of steady-state levels of c-Myc, ARMC6, BRAT1, and TIGD5. c-Myc was used as a positive control as it is a well known proteasome substrate. Densitometric quantification of the bands from the assay are shown in bar plots. For quantification, bands were first normalized to GAPDH as a loading control and subsequently normalized to zero hour, untreated samples (set to 1). CHX = cycloheximide.

The online version of this article includes the following source data and figure supplement(s) for figure 4:

*Figure 4 continued on next page*

*Figure 4 continued*

**Figure supplement 1.** Validation of PSMA4-miniTurbo cell line and application of ProteasomeID for detecting endogenous and PROTAC-induced proteasome substrates.

**Figure supplement 1—source data 1.** Raw unedited gels for *Figure 4—figure supplement 1*.

**Figure supplement 1—source data 2.** Uncropped and labeled gels for *Figure 4—figure supplement 1*.

the *Col1a1* locus downstream of a tetracycline responsive element (TRE) in the D34 mouse embryonic stem cell line (*Dow et al., 2014*). This line carries a cassette encoding the rtTA3 transactivator and the fluorescent protein mKate on the Rosa 26 locus under the control of a CAG promoter. Importantly, a LoxP-stop-LoxP cassette is present between the CAG promoter and the rtTA3 and mKate expressing cassette, enabling tissue-specific expression via crossing to specific CRE lines. The engineered D34 line was used to generate a mouse line via blastocyst injection. For proof of concept, we crossed the TRE-Psma4-miniTurbo;*Rosa26*-CAGs-RIK line with a CMV-Cre line that expresses constitutively the CRE recombinase in all tissues (*Nagy, 2000*). The obtained TRE-Psma4-miniTurbo;*Rosa26*-CAGs-RIK line was back crossed to C57BL/6 J to remove the CMV-Cre allele. The obtained mouse line constitutively expresses the rtTA3 transactivator, thereby enabling doxycycline inducible expression of the PSMA4-miniTurbo construct in all tissues.

We evaluated the induction of PSMA4-miniTurbo in different organs (kidney, liver, heart, skeletal muscle and brain) by feeding mice with doxycycline containing food for 10 or 31 days. We observed no significant changes in body weight nor any sign of suffering (*Figure 6b*). Immunoblot analysis confirmed expression of PSMA4-miniTurbo in all organs except the brain (*Figure 6—figure supplement 1a*). The locus used for PSMA4-miniTurbo expression (*Col1a1*) is known to be active in all the organs tested (*Lee, 2014*), however the limited penetration of food-administered doxycycline (*Gengenbacher et al., 2020*) limits the usability of our approach in the brain. The temporal dynamics of induction varied between organs. While 10 days of induction were sufficient to achieve PSMA4-miniTurbo protein levels comparable to endogenous PSMA4 in the kidney, other organs required 31 days. In most organs, the expression levels of PSMA4-miniTurbo were comparable to the one of endogenous PSMA4, with the exception of the skeletal muscle. In this organ, the levels of PSMA4-miniTurbo exceeded the endogenous PSMA4, suggesting the existence of a more prominent pool of not incorporated PSMA4-miniTurbo. Based on these observations, we decided to proceed with

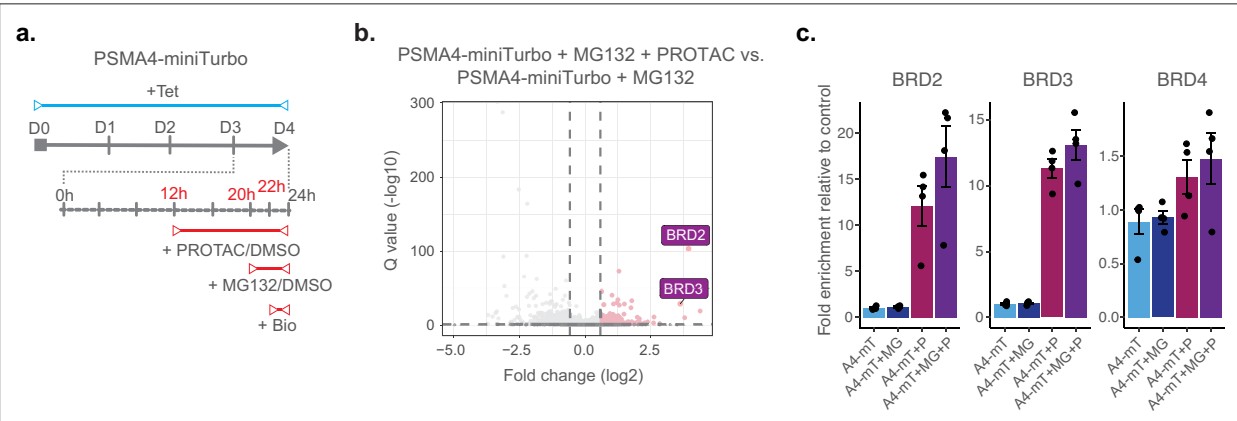

**Figure 5.** ProteasomeID identifies PROTAC-induced proteasome substrates. (**a**) Scheme of ProteasomeID workflow in HEK293T cells including proteasome inhibition by MG132 and treatment with PROTAC KB02-JQ1. The experimental design is analogous to the one depicted in (*Figure 4A*) with the additional PROTAC treatment achieved by addition of 10 µM KB02-JQ1 12 hr before cell harvesting. D: day; h: hour; Tet: tetracycline; Bio: biotin. (**b**) Volcano plot of proteins enriched by streptavidin pull-down and analyzed by DIA mass spectrometry from PSMA4-miniTurbo cells treated with KB02-JQ1 PROTAC molecule (P) and PSMA4-miniTurbo cells treated with both PROTAC molecule (P) and MG132 proteasome inhibitor (I). Cut offs for enriched proteins: log2 fold change >1 and Q value <0.05. n=4, biological replicates. Enrichment of BRD containing proteins is highlighted in violet boxes. (**c**) Bar plots comparing the levels of BRD-containing proteins following streptavidin enrichment from PSMA4-miniTurbo expressing cells exposed to the proteasome inhibitor MG132 and/or the PROTAC KB02-JQ1. mT: miniTurbo control cell line; A4-mT: PSMA4-miniTurbo cell line; I: proteasome inhibition by MG132; P: PROTAC (KB02-JQ1). Data are shown as mean ± standard error from n=4 biological replicates.

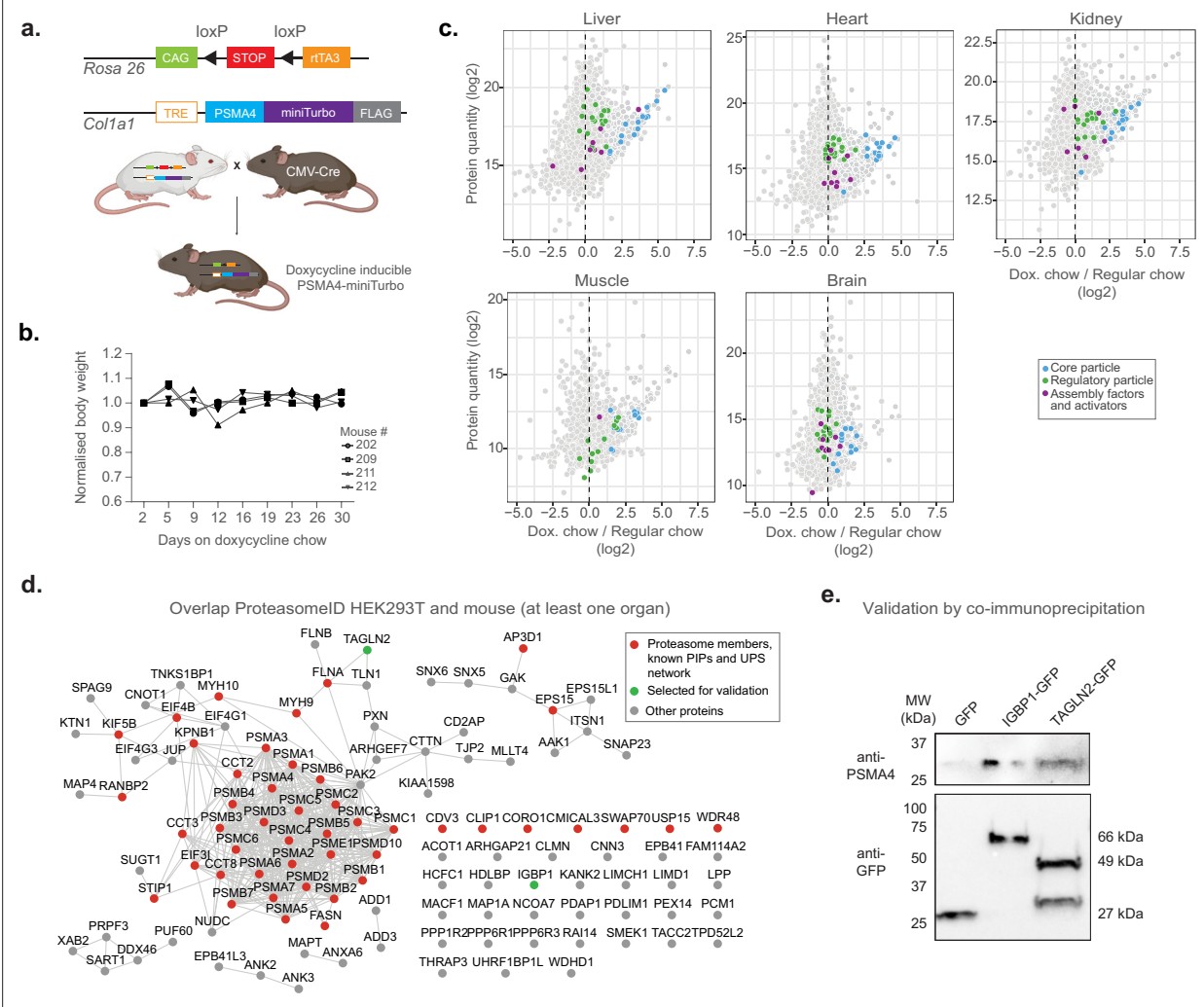

**Figure 6.** Establishment of a mouse model for in vivo ProteasomeID. (**a**) Design of a mouse model for ProteasomeID. The lox-STOP-lox cassette was excised from the *Rosa26* locus by crossing with a mouse line expressing the Cre recombinase under the control of an ubiquitous CMV promoter (**Nagy, 2000**). CAG: CAG promoter (**Miyazaki et al., 1989**), TRE: tetracycline-regulated element; rtTA3: reverse tetracycline-dependent transactivator A3 (**Dow et al., 2014**). Panel created with BioRender.com, and published using a CC BY-NC-ND license with permission. (**b**) Bodyweight curves of the experimental animals. The body weight for each mouse was normalized to its value at day 1 of the experiment (set to 1). (**c**) MA plots of proteins enriched by streptavidin pull-down and analyzed by DIA mass spectrometry from different mouse organs. Highlighted in color are proteasome members, assembly factors and activators, and other proteins significantly enriched in ProteasomeID (FPR <0.05). n=4 mice per experimental group. (**d**) Network analysis of overlap of significantly enriched proteins in ProteasomeID from HEK293T cells expressing PSMA4-BirA* and mouse organs (significant in at least one organ). Nodes colored in red indicate proteasome members, PIPs or proteins belonging to the UPS networks. The proteins selected for validation by co-immunoprecipitation are highlighted in green. Edges represent high confidence (>0.7) protein-protein interactions derived from the STRING database (**Snel et al., 2000**). (**e**) Cells expressing either GFP, IGBP1-GFP or TAGLN2-GFP were used for co-immunoprecipitation using GFP-trap. The elutions from GFP-trap were analyzed by immunoblot using antibodies against PSMA4 or GFP.

The online version of this article includes the following source data and figure supplement(s) for figure 6:

**Source data 1.** Raw unedited gels for *Figure 6*.

**Source data 2.** Uncropped and labeled gels for *Figure 6*.

**Figure supplement 1.** ProteasomeID mouse model validation and optimization.

**Figure supplement 1—source data 1.** Raw unedited gels for *Figure 6—figure supplement 1*.

**Figure supplement 1—source data 2.** Uncropped and labeled gels for *Figure 6—figure supplement 1*.

31 days of induction and confirmed expression of PSMA4-miniTurbo in the tested organs using anti-FLAG immunohistochemistry (*Figure 6—figure supplement 1b*).

In vivo BioID requires supplementation of exogenous biotin. We opted for subcutaneous biotin injection based on previous studies *Uezu et al., 2016*; *Uezu and Soderling, 2008*; *Spence et al., 2019*, and compared 3 x with 7 x daily biotin injections by assessing the recovery of proteasome subunits by LC-MS/MS upon streptavidin enrichment from liver and kidney lysates. We did not observe major quantitative differences in the enrichment of proteasome subunits relatively to control lysates from organs of mice that were not fed with doxycycline (*Figure 6—figure supplement 1c*). Therefore, we concluded that sufficient biotinylation of target proteins by ProteasomeID can be achieved with 3 x daily biotin injections.

Using the chosen conditions (31 days of doxycycline induction and 3 x biotin injections), we enriched biotinylated proteins from all organs except the brain and analyzed them by LC-MS/MS (*Figure 6—figure supplement 1d*). We revealed successful enrichment of proteasome components and known interacting proteins in all the organs tested (*Figure 6c* and *Supplementary file 5*). We then compared the ProteasomeID enriched proteins identified in cultured cells and mice, and identified 116 proteins shared between HEK293T and at least one mouse organ (*Figure 6d*). Forty-six of these proteins included proteasome subunits, PIPs or members of the UPS network. In addition, we detected 70 potential novel candidate proteasome interacting proteins that were consistently identified both in vitro and in vivo (*Supplementary file 5*). We selected two of these proteins for validation by co-immunoprecipitation (*Figure 6e*). We expressed immunoglobulin-binding protein 1 (IGBP1) and transgelin 2 (TAGLN2) as GFP-fusions in HEK293T cells, and performed pull downs using GFP-traps. Immunoblot analysis of GFP-trap eluates confirmed co-immunoprecipitation of these proteins and endogenous PSMA4. Together these data demonstrate that known and novel proteasome interacting proteins can be identified from mouse organs using ProteasomeID.

## Discussion

Our study demonstrates that proximity labeling coupled to mass spectrometry can be used to detect protein-protein interactions of the proteasome both in vitro and in vivo. Comparison to previous studies that used complementary biochemical approaches showed that ProteasomeID can retrieve a broad range of proteasome-interacting proteins in a single experiment. These include all the core proteasome subunits, activators, as well as assembly factors and components of the UPS, some of which have not been detected in previous studies. We demonstrated that ProteasomeID can be successfully implemented in mice allowing us to quantify proteasome interacting proteins in multiple mouse organs. The expression of tagged proteasomes did not negatively impact animal well-being. However, we noted that the relative levels of tagged proteasomes varied between organs, limiting the efficiency of ProteasomeID, especially in the brain. In the future, crossing the ProteasomeID mouse with lines expressing the Cre recombinase under the control of specific drivers will enable the study of proteasome composition and interactions with cell type resolution in vivo. However, for rare cell populations, pooling organs for multiple mice might be required to achieve sufficient input material for ProteasomeID. The ProteasomeID mouse might enable future mechanistic studies on the involvement of proteasomes in malignant, inflammatory, and metabolic diseases by means of crossing with established disease models. Recently, a mouse line has been reported enabling conditional expression of a 3xFLAG tagged version of the core proteasome subunit PSMA3/α7. Similarly to our findings, tagging of the proteasome at this location did not interfere with proteasome assembly or activity, and it enabled isolation of proteasomes specifically from neurons in different models of Alzheimer's disease (*Paradise et al., 2022*).

The combination of proximity labeling and proteasome inhibition enabled the identification of known proteasome activators and endogenous substrates of the proteasome, including low abundant proteins such as transcription factors that are typically challenging to quantify by bulk proteomic analysis. Previously, approaches based on proximity labeling and mass spectrometry have been used to detect substrates of ubiquitin ligases (*Yamanaka et al., 2022*; *Coyaud et al., 2015*) and other proteases, that is caspase-1 (*Jamilloux et al., 2018*). Our work demonstrates that a similar strategy can be implemented also for proteasomes. Importantly, the measured increases of protein abundance following proteasome inhibition were considerably more pronounced in ProteasomeID than total proteome analysis. This suggests that monitoring the proteasome proximal proteome might

provide higher sensitivity in the detection of proteasome substrates. Although we identified over 100 candidate substrates, the actual number of proteasome substrates is predicted to be much higher (*Wolf-Levy et al., 2018*). One possible explanation for this difference is that our study only examined a single time point (4 hr) following proteasome inhibition. This short duration may have excluded substrates with slower turnover rates. To obtain a more comprehensive understanding, future studies should consider investigating multiple time points, including longer durations, to capture a broader range of substrates, including those with slower turnover rates. We envisage that ProteasomeID could be used in the future to monitor how perturbation of cellular proteostasis by different proteotoxic stress, for example protein aggregate formation (*Guo et al., 2018*), influences the protein degradation landscape of the proteasome, or to monitor dynamic changes in proteasome interactions and substrates, for example during the cell cycle, cell differentiation, or cancer transformation.

Finally, we showed that ProteasomeID can directly identify known targets of PROTACs within cells. It is conceivable that the same strategy can be adapted to mouse models using the ProteasomeID mouse that we developed. This would enable assessing the efficacy of protein degraders in specific organs or cell types in vivo. In the future, quantification of the proteasome-associated proteome by ProteasomeID could greatly complement other existing approaches such as mass spectrometry analysis of proteolytic peptides (MAPP) (*Wolf-Levy et al., 2018*) to profile the proteasome degradation landscape in physiological and disease models, and to comprehensively assess the efficacy and specificity of protein degraders.

## Methods
### Mice

*Rosa26* mice (B6.Cg-*Col1a1*tm1(tetO-cDNA:Psma4)Mirim/J; B6.Cg-Gt(ROSA)26Sortm2 (CAG-rtTA3,-mKate2)Slowe/J) (*Dow et al., 2014*) were generated by Mirimus Inc (NY, USA). All animals were housed (2–5 mice per cage) at the Leibniz Institute on Aging - Fritz Lipmann Institute, in environmentally controlled, pathogen-free animal facility with a 12 hr light/ 12 hr dark cycle and fed ad libitum with a standard chow or with the doxycycline containing food (625 mg/kg of dry food) for 10 or 31 days. Animals used for the procedure were 2–4 months old. Biotin (24 mg/kg body weight) or PBS were administered subcutaneously, daily for the last 3 consecutive days of the regime. At the end of the regime, mice were euthanized with $CO_2$ in a $CO_2$ chamber (VetTech Solutions Ltd., AN045) and the organs isolated using scissors (FST, 14090–09) and forceps (FST, 11018–12). Isolated tissues to be used for mass spectrometry analysis were washed in PBS, weighted, snap-frozen in liquid nitrogen and stored at –80 °C. Isolated tissues to be used for immunofluorescence analysis were fixed in 4% formaldehyde, and embedded in paraffin for sectioning using a HistoCore Arcadia H and C (Leica). Four µm sections were cut using a microtome HM 340E (Thermo Fisher) and placed on microscope slides (Menzel, 041300). For subsequent immunofluorescence staining, sections were rehydrated through graded alcohols using a Autostainer XL (Leica) by two washes for 10 min in xylene followed by two washes in 100% ethanol for 3 min and 1 min each, 1 min wash in 95% ethanol, 1 min wash in 70% ethanol and 50% ethanol (all v/v in water). Next, slides were washed with PBS and following this usual protocol for immunofluorescence (see below) was used.

All the procedures were conducted with a protocol approved by animal experiment license NTP-ID 00040377-1-5 (FLI-20–010) in accordance with the guidelines of the 2010/63 EU directive as well as the instructions of GV SOLAS society.

## Cell culture and treatments

Flp-In T-REx 293 cells (Thermo Fisher Scientific, R78007), referred to as HEK293T, expressing PSMA4-BirA*, PSMC2-BirA*, BirA*-PSMD3, PSMA4-miniTurbo or miniTurbo were generated as described elsewhere (*Mackmull et al., 2017*). The parental cell line used was purchased directly from the vendor and no further authentication was performed. Cells were grown in DMEM high glucose 4.5 g/l supplemented with 10% (v/v) heat inactivated FBS, 2 mM L-Glutamine, 15 µg/ml Blasticidin and 100 µg/ml Hygromycin B. U-2 OS cells (ATCC, HTB-96; a kind gift from Pospiech lab, Leibniz Institute on Aging- Fritz Lipmann Institute) were grown in the same cell culture medium as FlpIn HEK293 T-REx cells, without antibiotics. All the cells were grown at 37 °C, 5% $CO_2$ and 95% humidity in a $CO_2$ incubator. The parental FlpIn T-REx 293 cell line was grown in presence of 100 µg/ml Zeocin and 15 µg/ml

Blasticidin. Upon generation of stable cell lines, Zeocin was replaced by 100 µg/ml Hygromycin B. All the cell lines were regularly tested for mycoplasma contamination.

For the BioID experiments, HEK293T lines were seeded at the density of approximately $1.6 \times 10^4$ cells/cm$^2$ and incubated for 24 hr to allow cell attachment to the culture dish. The expression of BirA* or miniTurbo fusion proteins were induced by a single addition of tetracycline stock (solved in ethanol) exposing the cells to its final concentration of 1 µg/µl in total for 4 days. Two h (miniTurbo lines) or 24 hr (BirA* lines) prior to cell harvesting, 50 µM biotin was added to the culture media. For identification of proteasome substrates, PSMA4-miniTurbo or miniTurbo expressing cells were treated with 20 µM MG132 for 4 hr and/or 10 µM KB02-JQ1 for 12 hr. Upon treatment, cells were washed 3 x times with PBS and harvested by trypsinization (0.05% trypsin incubated for 2 min at 37 °C). For each sample, a pellet corresponding to 20 million cells was collected and snap-frozen in liquid nitrogen.

## Immunoblot

Cell pellets were lysed in 50 mM HEPES pH 7.5, 5 mM EDTA, 150 mM NaCl, 1% (v/v) Triton X-100, prepared with phosphatase inhibitors and protease inhibitors for 30 min on ice. Lysates were cleared by centrifugation for 15 min at 21,000 x *g* at 4 °C, supernatants transferred to fresh tubes and mixed with loading buffer (1.5 M Tris pH 6.8, 20% SDS (w/v), 85% glycerin (w/v), 5% β-mercaptoethanol (v/v)). This was followed by denaturation for 5 min at 95 °C. Ten to 20 µg of sample, determined by use of EZQ Protein Quantitation Kit, was loaded on a 4–20% Mini-Protean TGX Gels (BIO-RAD) per lane and separated by SDS-PAGE. Proteins were transferred to a nitrocellulose membrane with a Trans-BlotTurbo Transfer Starter System (Bio-Rad, 170–4155). For high molecular weight samples (SYNJ1-BirA*, and BirA*-SYNJ1) a wet transfer method was used with Hoefer TE22 Mini Tank Blotting Unit (Thermo Fisher Scientific, 03-500-216), using a wet transfer buffer (25 mM Tris pH 8.3, 192 mM glycine, 15% (v/v) Methanol). Membranes were stained with PonceauS for 5 min on a shaker, washed and imaged on a Molecular Imager ChemiDoc XRS + Imaging system (Bio-Rad) and destained with TBST for 1 min at room temperature (RT). After incubation for 1 hr in blocking buffer (3% BSA (w/v), 25 mM Tris, 75 mM NaCl, 0.5% (v/v) Tween-20), membranes were incubated overnight at 4 °C with primary antibodies diluted in blocking buffer for FLAG M2 (1:1000), Streptavidin HRP (1:40,000), PSMA4 (1:250), SYNJ1 (1:250), β-actin (1:5000). This was followed by a 1 hr incubation with secondary antibodies dilution matching species conjugated with HRP (1:2000, anti-rabbit; 1:1500, anti-mouse, in 0.3% BSA in TBST (w/v)). Proteins were detected using the enhanced chemiluminescence detection kit (ECL) following the manufacturer instructions. Signals were acquired on the Molecular Imager ChemiDocXRS +Imaging system.

For immunoblots of anti-FLAG and Streptavidin-HRP on samples from PSMA4-BirA* and PSMC2-BirA*, the cells were lysed in RIPA buffer (150 mM NaCl, 1% Triton X-100 (v/v), 0.5% sodium deoxycholate (w/v), 0.1% SDS (w/v); 50 mM Tris, pH8) prepared with phosphatase inhibitors and protease inhibitors. Samples were incubated on ice for 10 min and lysates were prepared by sonication in Bioruptor Plus sonication device (Diagenode). The following steps were performed as indicated previously, but using a different buffer to check the efficiency of the proteins to the membrane (amido black solution 0.25% (w/v), naphthol blue black, 45% (v/v) methanol, 10% (v/v) acetic acid, in milliQ water). After the ECL reaction, membranes were visualized on a CL-XPosure Film (Thermo Fisher Scientific, 34090), using an Amersham Hypercassette Autoradiography Cassette (RPN11648).

## Proteasome activity assay

The proteasome activity assay (PAA) was performed using the 20 S proteasome activity assay kit (Millipore, APT280) following the manufacturer instructions. In short, cell pellets were thawed in ice-cold lysis buffer (50 mM HEPES pH 7.5, 5 mM EDTA, 150 mM NaCl, 1% (v/v) Triton X-100, 2 mM ATP) and left on ice for 30 min with short vortex steps every 10 min. Samples were centrifuged at 20,817 x *g*, for 15 min at 4 °C to remove any debris. For protein amount estimation the EZQ Protein Quantitation Kit was used. 50 µg of protein extract were incubated with fluorophore-linked peptide substrate (LLVY-7-amino-4-methylcoumarin, AMC) for 60 min at 37 °C. Proteasome activity was measured by quantification of fluorescent units from cleaved AMC at 380/460 nm using a microplate reader m1000 (Tecan).

## Cycloheximide chase assay

HEK293T cells were seeded in six-well plates (300 k cells per well). The cells were simultaneously incubated with 50 µg/ml cycloheximide for 0, 4, 8, and 12 hr in the presence or absence of 20 µM MG132 proteasome inhibitor. To induce expression of the construct (PSMA4-BirA*), cells were exposed to 1 µg/µl of tetracycline. Cells were then harvested as described above. Cell pellets were then prepared by the same method and lysis buffer described in Proteasome activity assay for western blot analysis. Ten µg of total proteins, determined by use of EZQ Protein Quantitation Kit, per lane were separated on SDS PAGE and analyzed by immunoblotting on anti-c-myc, anti-ARMC6, anti-BRAT1, anti-TIGD5 and anti-GAPDH antibodies. The intensity of the bands were analyzed with the Image Lab software (v6.0.1, Bio-Rad Laboratories, Inc).

## Native gel electrophoresis

Pellets of one million HEK293T cells containing an estimated 100 µg of total protein were collected. The pellets were then lysed in the same lysis buffer and in the same conditions as used for Proteasome activity assay. Following this, the samples were transferred into fresh tubes and 50 µg of each sample was subjected to native gel electrophoresis to reveal the various proteasome complexes (30 S, 26 S, 20 S) using a NativePAGE 3 to 12% Bis-Tris gel (Invitrogen, BN1001BOX) in XCell SureLock Mini-Cell system (Invitrogen). Appropriate running buffers for native page were used (NativePAGE Running Buffer Kit, Invitrogen, BN2007) with addition of providing 2 mM ATP in buffers coming into direct contact with the gel. The gel was run for 60 min at 4 °C with constant 150 V, followed by voltage increase to 250 V and continued to run for another 90 min. Proteins in native gels were transferred to nitrocellulose membranes for 2 hr at 28 V, 0.11 mA, at 4 °C, by using wet transfer buffer (48 mM Tris, 390 mM Glycine, 0.1% (w/v) SDS, 20% Methanol). The membranes were then blocked in 3% BSA, 0.5% TBST and immunoblotted with monoclonal antibodies against PSMA4 (1:250 dilution), FLAG M2 (1:1000 dilution) and PSMC2 (1:1000 dilution) overnight at 4 °C. Membranes were further incubated with horseradish peroxidase-conjugated secondary antibodies for 1 hr. Proteins were detected using the Pierce ECL Western Blotting Substrate (Thermo Scientific, 32106) and the intensity of the bands were analyzed by the software Image Lab (v6.0.1, Bio-Rad).

## GFP trap

For generation of cell lines transiently expressing either novel interactors fused to GFP or GFP control, plasmids were either bought or generated using the Gateway Technology (Invitrogen). Three 10 cm dishes of HEK293T (4 million cells/dish) were used for each candidate and prior to transfection, the medium was replaced with a transfection medium (DMEM with 2% FBS, without antibiotics). Cells were transfected with 5 µg plasmid DNA and 15 µg Polyethylenimine (PEI 25 K, Polysciences, 23966–100), both previously prepared in OptiMEM (Gibco, 11520386). The transfection medium was changed to DMEM with 10% FBS and 1% Pen-Strep 6 hr post-transfection and incubated for 48 hr at 3.5%, $CO_2$, 37 °C. Subsequently, cells were harvested by trypsinization and 20 million cell pellets were collected. For immunoprecipitation reactions, each pellet was lysed in 200 µl of lysis buffer (20 mM Tris pH 7.5, 150 mM NaCl, 1 mM EDTA, 0.3% Triton, 10% Glycerol) containing protease and phosphatase inhibitors. Tubes were placed on ice for 30 min with vortexing every 10 min, followed by brief sonication in a Bioruptor Plus for 5 cycles (30 s on/30 s off) at high setting and afterwards centrifuged at 20,000 x *g*, for 10 min at 4 °C. Lysate-supernatants were transferred to a pre-cooled tube and to each of them 300 µl Dilution buffer (10 mM Tris pH 7.5, 150 mM NaCl, 0.5 mM EDTA) was added. Twenty-five µl of GFP Trap beads (ChromoTek GFP-Trap Agarose, proteintech, gta) were equilibrated in 0.5 ml of ice-cold Dilution buffer and were spun down at 100 × *g* for 5–10 s at 4 °C. Beads were washed two more times with 500 µl Dilution buffer. In total, 2 mg of lysate-supernatant was added to equilibrated GFP Trap beads and were incubated for 1 hr, 4 °C with constant mixing. Tubes were spun at 100 × *g* for 5–10 s at 4 °C. GFP Trap beads were washed with 500 µl ice-cold Dilution buffer, followed by two time wash with Wash Buffer (10 mM Tris pH 7.5, 150 mM NaCl, 0.05% P40 Substitute, 0.5 mM EDTA). Eighty µl of 2 x SDS-sample buffer (120 mM Tris pH 6.8, 20% glycerol, 4% SDS, 0.04% bromophenol blue) was then added to the GFP Trap beads and boiled for 5 min at 95 °C. The beads were collected by centrifugation at 2500 × *g* for 2 min and SDS-PAGE was performed with the supernatant. Antibodies used for immunoblotting analysis: anti-PSMA4 (Novus Biologicals, NBP2-38754, 1:250), 2nd

Ab: anti-rabbit (Agilent Dako, P0448, 1:2000), anti-GFP (Santa Cruz, sc-9996, 1:1000), 2nd Ab: anti-mouse (Agilent Dako, P0447, 1:1500).

## Size-exclusion chromatography (SEC)

Pellets of 80 million HEK293T cells expressing PSMA4-BirA* were collected and snap frozen in liquid nitrogen. The pellets were resuspended in 2 ml lysis buffer (50 mM HEPES pH 6.8, 1 mM $MgCl_2$, 1 mM DTT, 20 mM NaCl, 5% glycerol, phosphatase inhibitors and protease inhibitors) and incubated 30 min on ice. Cell swelling and lysis was checked in 15 min intervals. Cell lysis was assisted by passage of the sample through a 27 G needle 12 times. Following this, the final concentration of NaCl was adjusted to 150 mM. The samples were then clarified by subsequent centrifugation steps as follows: (i) 500 x $g$ for 5 min at 4 °C, (ii) 1000 x $g$ for 13 min at 4 °C, and (iii) 100000 x $g$ for 30 min at 4 °C. The final supernatant was concentrated using 30 kDa cut-off spin filters (Merck Amicon Ultra −0.5 ml, centrifugal filters, UFC503096) to a final protein concentration of approximately 10 µg/µl measured by OD280, and further applied to size-exclusion chromatography.

SEC was performed using an ÄKTA avant (GE Äkta avant 25–1) system equipped with UV detection at 280 nm wavelength. A Yarra-SEC-4000 column (300×7.8 mm, pore size 500 Å, particle size 3 µm) was used with a SecurityGard cartridge GFC4000 4×3.0 mm ID as a guard column. Running conditions were 4 °C, a flow rate of 0.5 ml/min and run time of 40 min. The mobile phase contained 50 mM HEPES, pH 6.8, 1 mM $MgCl_2$, 1 mM DTT, 150 mM NaCl, and 5 mM ATP. A control sample (Phenomenex, ALO-3042) was injected prior to each sample to verify column performance. A total of 100 µl samples from 10 mg/ml lysate solution were injected, corresponding to 1 mg protein extract on column. Fractions (200 µl each) were collected along with the LC (liquid chromatography) separation directly in the SDS buffer, to a final concentration of 4%. Thirty-six fractions were further processed for LC-MS/MS analysis. Of these 36 fractions, the first and last two fractions were pooled.

## Preparation of SEC fractions for mass spectrometry analysis

The SEC fractions were further processed by addition of DTT (50 mM) in 100 mM HEPES at pH 8, boiled for 5 min at 95 °C, followed by sonication (Diagenode Bioruptor Plus) for 10 cycles (30 s on/60 s off) at 4 °C. The samples were then centrifuged at 3000 x $g$ for 5 min at RT, and the supernatant transferred to 2 ml tube. This was followed by alkylation with 20 mM iodoacetamide (IAA) for 30 min at RT in the dark. Protein amounts were confirmed by SDS–PAGE (4%). Protein samples in the collected fractions ranged from 10 to 100 µg. Proteins were precipitated overnight at −20 °C after addition of a 4×volume of ice-cold acetone. Thereafter, the samples were centrifuged at 20,800 x $g$ for 30 min at 4 °C and the supernatant carefully removed. Pellets were washed twice with 1 ml ice-cold 80% (v/v) acetone and then centrifuged with 20,800 x $g$ at 4 °C. The samples were air-dried before addition of 120 µl digestion buffer (3 M urea, 100 mM HEPES, pH8). Samples were resuspended by sonication (as above) and LysC was added at 1:100 (w/w) enzyme:protein ratio. The samples were then digested for 4 h at 37 °C (1000 x rpm for 1 hr, then 650 x rpm, Eppendorf ThermoMixerC). Samples were then diluted 1:1 with milliQ water, and trypsin added at the same enzyme to protein ratio and further digested overnight at 37 °C (650 x rpm). Consequently, digests were acidified by the addition of TFA to a final concentration of 2% (v/v) and then desalted with a Waters Oasis plate (HLB µElution Plate 30 µm, Waters, 186001828BA) with slow vacuum. Therefore, the columns were conditioned three times with 100 µl solvent B (80% (v/v) acetonitrile, 0.05% (v/v) formic acid) and equilibrated three times with 100 µl solvent A (0.05% (v/v) formic acid in Milli-Q water). The samples were loaded, washed 3 times with 100 µl solvent A, and then eluted with 50 µl solvent B. The eluates were dried in a vacuum concentrator.

## BioID affinity purification

Cell pellets were thawed on ice and resuspended in 4.75 ml BioID lysis buffer (50 mM Tris pH 7.5, 150 mM NaCl, 1 mM EDTA, 1 mM EGTA, 1% (v/v) Triton X-100, 1 mg/ml aprotinin, 0.5 mg/ml leupeptin, 250 U turbonuclease, 0.1% (w/v) SDS), followed by 1 hr incubation in the rotator mixer (STARLAB RM Multi-1) (15 rpm) at 4 °C to aid the lysis. Samples were then briefly sonicated in a Bioruptor Plus for five cycles (30 s on/30 s off) at high setting and afterwards centrifuged at 20,817 x $g$, for 30 min at 4 °C to remove any debris.

Mouse organs were thawed and transferred into Precellys lysing kit tubes (Bertin Instruments, 431–0170, Keramik-kit 1.4/2.8 mm, 2 ml (CKM)) containing 1 ml of PBS supplemented with 1 tab of complete, EDTA-free Protease Inhibitor per 50 ml. For homogenization, organs were shaken twice at 6000 x rpm for 30 s, centrifuged at 946 x $g$ at 4 °C for 5 min, and the resulting homogenate was transferred to a new tube. Based on the estimated protein content (5% of fresh tissue weight for liver, 8% for heart and kidney and 20% for muscle), homogenates corresponding to 4 mg protein were processed for further BioID affinity purification. This entailed cell lysis of the homogenates by means of BioID lysis buffer.

Streptavidin-coated Sepharose beads (Merck, GE17-5113-01) were acetylated by two successive treatments with 10 mM Sulfo-NHS-Acetate for 30 min at RT. The reaction was then quenched with 1 M Tris pH 7.5 (1:10 v/v) and the beads were washed three times with 1 x PBS and centrifuged at 2000 x $g$ for 1 min at RT. Cleared lysates were transferred to new tubes, 50 µl of acetylated beads added, and samples were incubated for 3 hr on the rotator (15 rpm) at 4 °C. This was followed by centrifugation at 2000 x $g$ for 5 min at 4 °C and removal of 4.5 ml of the supernatant from each sample. Remaining sample with the beads at the bottom was transferred to a Pierce Spin Column Snap Cap column (Thermo Fisher Scientific, 69725) and the tubes were additionally rinsed with a lysis buffer and added to the spin column. Beads were then washed on the column with a lysis buffer, followed by three washes with freshly prepared 50 mM ammonium bicarbonate (AmBic), with the pH adjusted to 8.3. The bottom of the columns were closed with a plug and beads transferred to fresh 2 ml tubes by means of 3x300 µl 50 mM AmBic, pH 8.3. The samples were then centrifuged at 2000 x $g$ for 5 min at 4 °C and the content of each tube was removed, leaving 200 µl in the tube. One µg of LysC was added and incubated at 37 °C for 16 hr shaking at 500 x rpm. The samples were then centrifuged at 2000 x $g$ for 5 min at room temperature and the content of the tubes were transferred to Pierce Spin Column Snap Cap columns. The digested peptides were eluted with two times 150 µl of freshly made 50 mM AmBic. To elute biotinylated peptides still bound to the beads, 150 µl of 80% ACN and 20% TFA was added, briefly mixed, and rapidly eluted. This elution step was repeated twice and the eluates merged. Following elution, 0.5 µg of trypsin was added to the AmBic elutions and digestion continued for an additional 3 hrwith mixing at 500 x rpm and 37 °C. Digested AmBic elutions were then dried down in a vacuum concentrator, resuspended in 200 µl 0.05% (v/v) formic acid in milliQ water and sonicated in a Bioruptor Plus (5 cycles with 1 min on and 30 s off with high intensity at 20 °C). ACN/TFA elutions were dried down in a vacuum concentrator until approximately 50 µl were left, and 50 µl of 200 mM HEPES pH 8.0 were added to the samples and pH adjusted to 7–9. 0.5 µg of trypsin were then added and digestion continued for an additional 3 hr with mixing at 500 x rpm at 37 °C. Digested peptides were acidified with 10% (v/v) trifluoroacetic to pH <3. Both AmBic and ACN/TFA elutions were desalted using Macro Spin Column C18 columns (Thermo Scientific, Pierce, 89873) following manufacturer's instructions and dried down in a vacuum concentrator.

## LC-MS/MS data acquisition

Prior to analysis, samples were reconstituted in mass spectrometry (MS) Buffer (5% acetonitrile, 95% Milli-Q water, with 0.1% formic acid) and spiked with iRT peptides. Peptides were separated in trap/elute mode using the nanoAcquity MClass Ultra-High Performance Liquid Chromatography system (UPLC) or nanoAcquity UPLC system (Waters) equipped with a trapping (nanoAcquity Symmetry C18, 5 µm, 180 µm×20 mm) and an analytical column (nanoAcquity BEH C18, 1.7 µm, 75 µm×250 mm). Solvent A was water with 0.1% formic acid, and solvent B was acetonitrile with 0.1% formic acid. One1 µl of the sample (~1 µg on column) was loaded with a constant flow of solvent A at 5 µl/min onto the trapping column. Trapping time was 6 min. Peptides were eluted via the analytical column with a constant flow of 0.3 µl/min. During the elution, the percentage of solvent B increased in a nonlinear fashion from 0–40% in 90 min (120 min for total proteome of mouse organs). Total run time was 115 min (145 min) including equilibration and conditioning. The LC was coupled to an Orbitrap Fusion Lumos (Thermo Fisher Scientific) using the Proxeon nanospray source or to an Orbitrap Q-Exactive HFX (Thermo Fisher Scientific) for BioID experiments from HEK293T cells, or to an Orbitrap Exploris 480 (Thermo Fisher Scientific) for BioID experiments combined with PROTAC treatment. The peptides were introduced into the mass spectrometer via a Pico-Tip Emitter 360 µm outer diameter ×20 µm inner diameter, 10 µm tip (New Objective) heated at 300 °C, and a spray voltage of 2.2 kV was applied. For data acquisition and processing of the raw data Tune version 2.1 and Xcalibur

4.1 (Orbitrap Fusion Lumos), Tune 2.9 and Xcalibur 4.0 (Orbitrap Q-Exactive HFX) and Tune 3.1 and Xcalibur 4.4 (Orbitrap Exploris 480) were employed.

*DDA (Data-dependent acquisition)*. SEC fractions, mouse BioID as well as BioID of PSMA4 and PSMC2 were analyzed using DpD (DDA plus DIA). Here, data from a subset of conditions were first acquired in DDA mode to contribute to a sample specific spectral library. Full scan MS spectra with mass range 375–1500 m/z (using quadrupole isolation) were acquired in profile mode in the Orbitrap with resolution of 60,000 FWHM. The filling time was set at a maximum of 50ms with a limitation of $2x10^5$ ions. The 'Top Speed' method was employed to take the maximum number of precursor ions (with an intensity threshold of $5x10^5$) from the full scan MS for fragmentation (using HCD collision energy, 30%) and quadrupole isolation (1.4 Da window) and measurement in the Orbitrap, with a cycle time of 3 s. The MIPS (monoisotopic precursor selection) peptide algorithm was employed. MS/MS data were acquired in centroid mode in the Orbitrap, with a resolution of 15,000 FWHM and a fixed first mass of 120 m/z. The filling time was set at a maximum of 22ms with a limitation of $1x10^5$ ions. Only multiply charged (2+ - 7+) precursor ions were selected for MS/MS. Dynamic exclusion was employed with maximum retention period of 15 s and relative mass window of 10 ppm. Isotopes were excluded.

*DIA (Data-independent acquisition)*. The DIA data acquisition was the same for both directDIA and DpD. Full scan mass spectrometry spectra with mass range 350–1650 m/z were acquired in profile mode in the Orbitrap with resolution of 120,000 FWHM. The default charge state was set to 3+. The filling time was set at a maximum of 60ms with a limitation of $3×10^6$ ions. DIA scans were acquired with 34 mass window segments of differing widths across the MS1 mass range. Higher collisional dissociation fragmentation (stepped normalized collision energy: 25, 27.5, and 30%) was applied and MS/MS spectra were acquired with a resolution of 30,000 FWHM with a fixed first mass of 200 m/z after accumulation of $3×10^6$ ions or after filling time of 35ms (whichever occurred first). Data was acquired in profile mode.

## LC-MS/MS data analysis

DpD (DDA plus DIA) libraries were created by searching both the DDA runs and the DIA runs using Spectronaut Pulsar (v 13–15, Biognosys). The data were searched against species specific protein databases (*Homo sapiens*, reviewed entry only (16,747 entries), release 2016_01 or *Mus musculus*, entry only (20,186), release 2016_01, respectively) with a list of common contaminants appended. The data were searched with the following modifications: carbamidomethyl (C) as fixed modification, and oxidation (M), acetyl (protein N-term), and biotin (K) as variable modifications. A maximum of 2 missed cleavages was allowed. The library search was set to 1% false discovery rate (FDR) at both protein and peptide levels. This library contained 79,732 precursors, corresponding to 4730 protein groups for SEC fractions, 77,401 precursor, corresponding to 5125 protein groups for BioID on PSMA4 and PSMC2 using Spectronaut protein inference. All other BioID experiments were processed using the directDIA pipeline in Spectronaut Professional (v.13–17). The data were searched against a species specific (*Mus musculus* and *Homo sapiens*, as described above) with a list of common contaminants appended. BGS factory settings were used with the exception of: variable modifications = acetyl (protein N-term), biotin (K), oxidation (M).

SEC-MS experiments were processed using Spectronaut v.13 with default settings except: Proteotypicity Filter = Only Protein Group Specific; Major Group Quantity = Median peptide quantity; Major Group Top N=OFF; Minor Group Quantity = Median precursor quantity; Minor Group Top N=OFF; Data Filtering = Qvalue sparse; Imputing Strategy = No imputing; Cross run normalization = OFF.

PSMA4 and PSMC2 BioID experiments were processed using Spectronaut v.13 with default settings except: Proteotypicity Filter = Only Protein Group Specific; Major Group Quantity = Median peptide quantity; Major Group Top N=OFF; Minor Group Quantity = Median precursor quantity; Minor Group Top N=OFF; Data Filtering = Qvalue percentile (0.5); Imputing Strategy = No imputing; Normalization Strategy = Global Normalization; Normalize on = Median; Row Selection = Qvalue sparse.

Mouse BioID, PSMD3 BioID and BioID experiments combined with PROTAC treatment were processed using Spectronaut v.15, v.17 and v18 with default settings except: Proteotypicity Filter = Only Protein Group Specific; Major Group Quantity = Median peptide quantity; Major Group Top N=OFF; Minor Group Quantity = Median precursor quantity; Minor Group Top N=OFF; Data Filtering

= Qvalue percentile (0.2); Imputing Strategy = Global imputing; Normalization Strategy = Global Normalization; Normalize on = Median; Row Selection = Qvalue complete.

For all the BioID experiments, differential abundance testing was performed in Spectronaut using a paired t-test between replicates. p values were corrected for multiple testing multiple testing correction with the method described by *Storey, 2002*. The candidates and protein report tables were exported from Spectronaut and used for further data analysis using R and RStudio server.

## Logistic regression classifier for detecting ProteasomeID-enriched proteins

To identify ProteasomeID-enriched proteins, we trained a logistic regression binary classifier. The classifier was trained using known proteasome members as positive class, and mitochondrial matrix proteins as negative. Mitochondrial matrix proteins are not expected to interact directly with the proteasome under homeostatic conditions. To distinguish between these two classes, we performed prediction using an enrichment score derived from multiplying the average log2 ratio and the negative logarithm of the q-value obtained from a differential protein abundance analysis performed against the BirA* control line. Before analysis, any missing data points were removed from the dataset. To assess the performance of the binary classifier, and optimize its parameters, a 10-fold cross-validation approach was adopted. The dataset was randomly partitioned into ten subsets, with nine subsets used for training and one subset for validation in each iteration. This process was repeated thirty times, and results were averaged using the mean value to ensure the robustness of the results. Logistic regression was employed as the classification method using the caret package in R *Kuhn, 2008*. To determine an optimal threshold for classification, the false positive rate (FPR) was set at 0.05. The threshold yielding an FPR closest to the target value of 0.05 was selected as the final classification threshold. Following model training and threshold selection, the classifier was applied to predict the class labels of additional proteins not used in the training process. The enrichment score and class labels for the new data were provided as input to the trained model. All statistical analyses were performed using R version 4.1.3. The pROC (*Robin et al., 2011*) and caret (*Kuhn, 2008*) packages were employed for ROC analysis and logistic regression, respectively.

## Immunofluorescence

Cells were grown on autoclaved coverslips (Carl Roth, YX03.1), coated with Poly-D-Lysine. Coverslips were placed individually in 12-well plates (Lab solute, 7696791), and 25 k cells were seeded per well. Cells were washed three times with 1 x PBS, fixed in 4% formaldehyde (v/v) in PBS for 10 min at RT, washed 3x5 min with 1 x PBS and permeabilized with permeabilization buffer (0.7% Triton X-100, in 1 x PBS) at room temperature for 15 min. Washing with PBS was repeated 2x5 min and samples were incubated with blocking solution (10% (w/v) BSA, 10% (v/v) Triton X-100, 5% (v/v) goat serum) for 10 min at RT. The coverslips were incubated with primary antibody anti-FLAGM2 (1:100, Sigma Aldrich, mouse F3165) or anti-FLAG (1:500, Sigma-Aldrich, mouse, F1804)at 4 °C overnight. After washing 3x5 min with PBS/PBST (first with PBS, second with PBS + 0.2% (v/v) Tween 20, third with PBS) the secondary fluorescence-labeled antibody (goat anti-mouse IgG (H+L) - Cyanine5, 1:400 in blocking solution or goat anti-mouse IgG g1 Alexa Fluor 488, 1:1000 in blocking solution) and fluorescently labeled streptavidin, 1:2000 in blocking solution were incubated for 30 min at 37 °C. After 3x5 min with PBS/PBST (first with PBS, second with PBS + 0.2% (v/v) Tween 20, third with PBS), nuclei were stained with DAPI (4',6-Diamidino-2-Phenylindole, Dihydrochloride, 0.02 µg/µl in PBS) at RT for 10 min and washed again with PBS 2x5 min. Frozen sections were mounted in Permafluor mounting medium using glass slides (041300, Menzel) and dried at room temperature overnight. All samples were stored at 4 °C in the dark until further analysis by microscopy. Immunofluorescence microscopy was performed with an Axio Imager (Z2 using a Plan-Apochromat 63 x / 0.8 M27 Objective) and analyzed with the software Zen 2 Blue Edition (Carl Zeiss Microscopy GmbH).

## Immunohistochemistry

Five µm sections were deparaffinized by xylene twice 5 min each and rehydrated by 100%, 90%, 70% ethanol for 5 min each and washed with tap water for 10 min. Sections were treated with antigen unmasking solution sodium citrate buffer (Vector, H-3300) in the microwave at 800 W (3 min) followed by 400 W (10 min) for antigen retrieval. Sections were cooled at room temperature and washed three

times with PBS, 5 mins each. Sections were blocked for the endogenous peroxidase activity by 0.03% $H_2O_2$ in methanol for 30 min at RT. Sections were rinsed with PBS three times and blocked with 5% BSA for 1 hr at RT. Excess serum was tipped off and sections were incubated with primary antibody (anti-FLAG, Sigma F7425, 1:100 diluted in 1% BSA), overnight in a humid chamber at 4 °C. The next day sections were brought to room temperature and washed three times with PBS, 5 min each. Sections were incubated with biotinylated rabbit antibody diluted in 1% BSA/PBS +0.01% Tween20 for 1 hr. Sections were washed three times with PBS, 5 min each and incubated with a mixture from VECTA-STAIN Elite ABC HRP Kit (PK-6100) for 30 min. It was prepared at least 20 min prior to use, one drop of solution A and one drop of solution B were mixed into 2.5 ml of ABC dilution buffer, vortexed and used. Sections were rinsed three times with PBS, 5 min each and developed by ImmPACT NovaRED Peroxidase (HRP) Substrate (SK-4800) according to manufacturer's instructions until the reddish brown color visibly appeared while examining the sections under the microscope. The reaction was stopped by immersing the sections in the water for 5 min. Sections were counterstained with hematoxylin for 30 s, rinsed in tap water, and dehydrated with 70%, 90%, and 100% ethanol 30 s each and with xylene for 1 min. Slides were mounted with xylene based mounting medium, air dried and stored in a cold and dry place for further analysis. Images were taken with an Axio Imager M2 microscope (Zeiss) equipped with an AxioCam MRc5 (Zeiss), with a 20 x objective.

## Molecular visualization and structure analysis

For visualization of proteasome complexes UCSF Chimera program (version 1.13.1) was used. The three-dimensional structural data of macromolecular complexes of proteasome were downloaded from the Protein Data Bank (PDB) database (5T0C). For the analysis of the enrichment of proteasome subunits in BioID protocol, data sets with fold change information were used and filtered the following way: q value <0.05, number of identified unique peptides per protein ≥2. The intensity of the proteasome subunit coloring used was directly dependent on the fold change of the identified subunit in the BioID affinity purification.

## Material availability

Plasmids generated in this study are available via AddGene. The cell lines and mouse strain are available from the authors upon request.

# Acknowledgements

The authors gratefully acknowledge support from the FLI Core Facilities Proteomics, Imaging, Functional Genomics, Histology & Electron Microscopy and the Mouse Facility. The authors acknowledge Anja Baar, Christoph Kaether for support with animal experiments. AO acknowledges funding from the DFG (RTG2155 ProMoAge), the Else Kröner Fresenius Stiftung (award number: 2019_A79), the Deutsches Zentrum für Herz-Kreislaufforschung (award number: 81X2800193), the Fritz-Thyssen foundation (award number: 10.20.1.022MN), the Chan Zuckerberg Initiative Neurodegeneration Challenge Network (award numbers: 2020–221617, 2021–230967 and 2022–250618), and the NCL Stiftung. The FLI is a member of the Leibniz Association and is financially supported by the Federal Government of Germany and the State of Thuringia.

# Additional information

## Competing interests

Aleksandar Bartolome, Julia C Heiby, Joanna M Kirkpatrick, Therese Dau, Alessandro Ori: inventors of the patent application WO 2024/013381A1. The other authors declare that no competing interests exist.

## Funding

| Funder | Grant reference number | Author |
| --- | --- | --- |
| Deutsche Forschungsgemeinschaft | RTG2155 | Alessandro Ori |
| Else Kröner-Fresenius-Stiftung | 2019_A79 | Alessandro Ori |
| Deutsches Zentrum für Herz-Kreislaufforschung | 81X2800193 | Alessandro Ori |
| Fritz Thyssen Stiftung | 10.20.1.022MN | Alessandro Ori |
| Chan Zuckerberg Initiative | 2020-221617 | Alessandro Ori |
| Chan Zuckerberg Initiative | 2021-230967 | Alessandro Ori |
| Chan Zuckerberg Initiative | 2022-250618 | Alessandro Ori |
| NCL Foundation | | Alessandro Ori |

The funders had no role in study design, data collection and interpretation, or the decision to submit the work for publication.

## Author contributions

Aleksandar Bartolome, Conceptualization, Formal analysis, Investigation, Visualization, Methodology, Writing – original draft; Julia C Heiby, Conceptualization, Supervision, Writing – review and editing; Domenico Di Fraia, Formal analysis, Visualization; Ivonne Heinze, Ellen Spaeth, Omid Omrani, Alberto Minetti, Maleen Hofmann, Investigation; Hannah Knaudt, Investigation, Visualization; Joanna M Kirkpatrick, Conceptualization, Investigation, Methodology, Writing – review and editing; Therese Dau, Conceptualization, Supervision, Project administration, Writing – review and editing; Alessandro Ori, Conceptualization, Resources, Supervision, Visualization, Writing – original draft, Project administration

## Author ORCIDs

Ellen Spaeth ⓘ https://orcid.org/0000-0002-3851-3931
Therese Dau ⓘ https://orcid.org/0000-0003-0251-9490
Alessandro Ori ⓘ https://orcid.org/0000-0002-3046-0871

## Ethics

All the procedures were conducted with a protocol approved by animal experiment license NTP-ID 00040377-1-5 (FLI-20-010) in accordance with the guidelines of the 2010/63 EU directive as well as the instructions of GV SOLAS society.

Reviewer #2 (Public Review): https://doi.org/10.7554/eLife.93256.3.sa1
Reviewer #3 (Public Review): https://doi.org/10.7554/eLife.93256.3.sa2
Author response https://doi.org/10.7554/eLife.93256.3.sa3

---

# Additional files

## Supplementary files

• Supplementary file 1. BioID data from cell lines expressing different biotin ligase fusion proteins.

• Supplementary file 2. SEC-MS data and biotinylation sites identified by ProteasomeID. (Tab1) SEC-MS data from HEK293T cells expressing PSMA4-BirA*. (Tab2) Biotinylation sites on proteasome subunits identified by 1015 PSMA4-BirA*.

• Supplementary file 3. Lists of known Proteasome Interacting Proteins (PIPs) and proteins enriched by ProteasomeID. (Tab1) List of PIPs from Bousquet et al., Fabre et al., Gedaki et al. (Tab2) List of PIPs from other studies. (Tab3) Protein groups enriched by ProteasomeID for PSMA4-BirA* dataset by the classifier algorithm and overlap with previous studies. (Tab4) GO enrichment candidate novel proteasome interactors.

• Supplementary file 4. ProteasomeID data following MG132 and KB02-JQ1 treatment. (Tab1) ProteasomeID data following treatment with MG132. (Tab2) Overlap between protein groups

enriched only upon MG132 inhibition and the ones whose level of ubiquitination is shown to increase upon MG132 inhibition in a previous study. (Tab3) ProteasomeID data following treatment with PROTAC KB02-JQ1.

• Supplementary file 5. ProteasomeID data from mouse organs. (Tab1-5) Protein groups enriched by ProteasomeID for mouse dataset. (Tab6) Overlap of ProteasomeID enriched proteins between HEK293T and mouse organs.

• MDAR checklist

### Data availability

Mass spectrometry proteomics data have been deposited to the ProteomeXchange Consortium via the PRIDE (*Perez-Riverol et al., 2019*) partner repository and they are accessible with the identifier: PXD032833 for the HEK293T ProteasomeID dataset for PSMA4 and PSMC2; PXD034874 for the HEK293T size exclusion chromatography dataset; PXD033008 for the MG132 ProteasomeID dataset; PXD032976 for the PROTAC ProteasomeID dataset; PXD034965 for the MG132 whole cell proteome dataset; Mass spectrometry proteomics data have been deposited to ProteomeXchange Consortium via the MassIVE partner repository and they are accessible with the identifier: MSV000092396 for the mouse ProteasomeID dataset MSV000092407 for the HEK293T ProteasomeID for PSMD3. The computer code used for the analysis presented in this manuscript is available at GitLab (copy archived at *Ori and Di Fraia, 2024*).

The following datasets were generated:

| Author(s) | Year | Dataset title | Dataset URL | Database and Identifier |
|---|---|---|---|---|
| Heiby JC, Fraia Di D, Heinze I, Knaudt H, Späth E, Omrani O, Minetti A, Hofmann M, Kirkpatrick JM, Dau T, Ori A, Bartolome A | 2024 | HEK293T ProteasomeID dataset for PSMA4 and PSMC2 | https://www.ebi.ac.uk/pride/archive/projects/PXD032833 | PRIDE, PXD032833 |
| Bartolome A, Heiby JC, Fraia Di D, Heinze I, Knaudt H, Späth E, Minetti A, Hofmann M, Kirkpatrick JM, Dau T, Ori A, Omrani O | 2024 | HEK293T size exclusion chromatography dataset | https://www.ebi.ac.uk/pride/archive/projects/PXD034874 | PRIDE, PXD034874 |
| Bartolome A, Heiby JC, Fraia Di D, Heinze I, Knaudt H, Späth E, Omrani O, Minetti A, Hofmann M, Kirkpatrick JM, Dau T, Ori A | 2024 | MG132 ProteasomeID dataset | https://www.ebi.ac.uk/pride/archive/projects/PXD033008 | PRIDE, PXD033008 |
| Bartolome A, Heiby JC, Fraia Di D, Heinze I, Knaudt H, Späth E, Minetti A, Hofmann M, Kirkpatrick JM, Dau T, Ori A, Omrani O | 2024 | PROTAC ProteasomeID dataset | https://www.ebi.ac.uk/pride/archive/projects/PXD032976 | PRIDE, PXD032976 |
| Bartolome A, Heiby JC, Fraia Di D, Heinze I, Knaudt H, Späth E, Minetti A, Hofmann M, Kirkpatrick JM, Dau T, Ori A, Omrani O | 2024 | MG132 whole cell proteome dataset | https://www.ebi.ac.uk/pride/archive/projects/PXD034965 | PRIDE, PXD034965 |

*Continued on next page*

*Continued*

| Author(s) | Year | Dataset title | Dataset URL | Database and Identifier |
|---|---|---|---|---|
| Bartolome A, Heiby JC, Fraia Di D, Heinze I, Knaudt H, Späth E, Omrani O, Minetti A, Hofmann M, Kirkpatrick JM, Dau T, Ori A | 2024 | Mouse ProteasomeID dataset | https://massive.ucsd.edu/ProteoSAFe/dataset.jsp?accession=MSV000092396 | MassIVE, MSV000092396 |
| Bartolome A, Heiby JC, Fraia Di D, Heinze I, Knaudt H, Späth E, Minetti A, Hofmann M, Kirkpatrick JM, Dau T, Ori A, Omrani O | 2024 | HEK293T ProteasomeID for PSMD3 | https://massive.ucsd.edu/ProteoSAFe/dataset.jsp?accession=MSV000092407 | MassIVE, MSV000092407 |

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

# Appendix 1

**Appendix 1—key resources table**

| Reagent type (species) or resource | Designation | Source or reference | Identifiers | Additional information |
|---|---|---|---|---|
| gene (*Homo sapiens*) | *psma4* | Uniprot | P25789 | |
| gene (*H. sapiens*) | *psma4* | Uniprot | Q9R1P0 | |
| Strain (*Mus musculus*) | *Rosa26*-CAGs-RIK mice | ***Nagy, 2000*** | | |
| Strain (*M. musculus*) | CMV-Cre line | ***Nagy, 2000*** | | |
| Strain (*M. musculus*) | *Rosa26* mice (B6.Cg-*Col1a1*tm1(tetO-cDNA:Psma4)Mirim/J; B6.Cg-Gt(ROSA)26Sortm2 (CAG-rtTA3,-mKate2) Slowe/J) | ***Dow et al., 2014***, Mirimus Inc (NY, USA) | | |
| Cell line (*H. sapiens*) | Flp-In T-REx 293 cells | Invitrogen | R78007 | |
| Transfected construct (*H. sapiens*) | PSMA4_BirA_Cterm | Addgene ID: 221513 | | |
| Transfected construct (*H. sapiens*) | PSMD3_BirA_Nterm | Addgene ID: 221526 | | |
| Transfected construct (*H. sapiens*) | Igbp1-GFP | Addgene ID: 221529 | | |
| Transfected construct (*H. sapiens*) | Tagln2-GFP | Addgene ID: 221530 | | |
| Transfected construct (*H. sapiens*) | PSMC2_BirA_Cterm | This paper (***Figure 1—figure supplement 3***) | | |
| Transfected construct (*H. sapiens*) | PSMA4_miniTurbo_Cterm | Addgene ID: 222600 | | |
| Antibody | anti-FLAG M2 (mouse monoclonal) | Sigma Aldrich | F3165 | 1:1000 |
| Antibody | anti-FLAG M2 (mouse monoclonal) | Sigma Aldrich | F1804 | 1:500 |
| Antibody | anti-PSMA4 (rabbit polyclonal) | NOVUS biologicals | NBP2-38754 | 1:250 |
| Antibody | anti-β-actin (mouse monoclonal) | Sigma Aldrich | A5441 | 1:5000 |
| Antibody | anti-PSMC2 (rabbit polyclonal) | Proteintech | 14905–1-AP | 1:1000 |
| Antibody | anti-c-Myc (rabbit monoclonal) | Abcam | ab32072 | 1:1000 |
| Antibody | anti-Ubi-K48 (rabbit monoclonal) | Millipore | 05–1307 | 1:1000 |
| Antibody | anti-ARMC6 (rabbit polyclonal) | Sigma Aldrich | HPA041420 | 1:2000 |
| Antibody | anti-BRAT1 (rabbit monoclonal) | Abcam | ab181855 | 1:10000 |
| Antibody | anti-GAPDH (mouse monoclonal) | Santa Cruz | sc-365062 | 1:200 |
| Antibody | anti-rabbit HRP-conjugated (goat polyclonal) | Dako | P0448 | 1:2000 |
| Antibody | anti-mouse HRP-conjugated (goat polyclonal) | Dako | P0447 | 1:1500 |
| Antibody | anti-FLAG (rabbit polyclonal) | Sigma Aldrich | F7425 | 1:100 |
| Antibody | anti-mouse-Cyanine5 (goat polyclonal) | Thermo Fisher Scientific | A10524 | 1:400 |
| Antibody | anti-mouse Alexa Fluor 488 (goat polyclonal) | Invitrogen | A21121 | 1:1000 |
| Antibody | anti-Proteasome 20 S alpha 1+2 + 3+5 + 6+7 (mouse monoclonal) | Abcam | ab22674 | 1:200 |
| Antibody | anti-GFP (mouse monoclonal) | Santa Cruz | sc-9996 | 1:1000 |
| Antibody | anti-BirA (mouse monoclonal) | Novus biologicals | NBP2-59939 | 1:500 |

*Appendix 1 Continued on next page*

*Appendix 1 Continued*

| Reagent type (species) or resource | Designation | Source or reference | Identifiers | Additional information |
|---|---|---|---|---|
| Antibody | anti-TIGD5 (rabbit polyclonal) | Proteintech | 13644–1-AP | 1:1000 |
| Peptide, recombinant protein | Streptavidin HRP | Abcam | ab7403 | 1:40000 |
| Peptide, recombinant protein | Streptavidin Alexa Fluor 568 | Invitrogen | S11226 | 1:2000 |
| Peptide, recombinant protein | Aprotinin | Carl Roth | A162.3 | |
| Peptide, recombinant protein | Leupeptin | Carl Roth | CN33.2 | |
| Peptide, recombinant protein | Trypsin | Promega | V511 | |
| Peptide, recombinant protein | Trypsin-EDTA | Thermo Fisher Scientific | 25300–062 | |
| Peptide, recombinant protein | LysC | Wako | 125–05061 | |
| Peptide, recombinant protein | Phusion High-Fidelity DNA Polymerase | NEB | M0530S | |
| Peptide, recombinant protein | Turbonuclease | MoBiTec GmbH | GE-NUC10700-01 | |
| Peptide, recombinant protein | KB02-JQ1 | MedChemExpress | HY-129917 | |
| Peptide, recombinant protein | Bovine serum albumin | Carl Roth | 3737.3 | |
| Commercial assay or kit | EZQ Protein Quantitation Kit | Invitrogen | R33200 | |
| Commercial assay or kit | Pierce ECL Western Blotting Substrate | Thermo Fisher Scientific | 32106 | |
| Commercial assay or kit | 20 S proteasome activity assay kit | Millipore | APT280 | |
| Commercial assay or kit | NativePAGE Running Buffer Kit | Invitrogen | BN2007 | |
| Commercial assay or kit | Precellys lysing kit | Bertin Instruments | 431–0170 | Keramik-kit 1.4/2.8 mm, 2 ml |
| Commercial assay or kit | VECTASTAIN Elite ABC HRP Kit | VectorLabs | PK-6100 | |
| Commercial assay or kit | Duolink In Situ PLA Probe Anti-Mouse MINUS | Sigma Aldrich | DUO92004 | |
| Commercial assay or kit | Duolink In Situ PLA Probe Anti-Rabbit PLUS | Sigma Aldrich | DUO92002 | |
| Commercial assay or kit | Duolink In Situ Detection Reagents Red | Sigma Aldrich | DUO92008 | |
| Commercial assay or kit | Duolink In Situ Wash Buffers, Fluorescence | Sigma Aldrich | DUO82049 | |
| Commercial assay or kit | iRT kit | Biognosys | Ki-3002–1 | |
| Chemical compound, drug | Biotin | Sigma Aldrich | B4501 | |

*Appendix 1 Continued on next page*

*Appendix 1 Continued*

| Reagent type (species) or resource | Designation | Source or reference | Identifiers | Additional information |
|---|---|---|---|---|
| Chemical compound, drug | D(+)-biotin | Sigma Aldrich | 3822.1 | |
| Chemical compound, drug | Duolink In Situ Mounting Medium with DAPI | Sigma Aldrich | DUO82040 | |
| Chemical compound, drug | L-Glutamine | Sigma Aldrich | G7513 | |
| Chemical compound, drug | complete Mini EDTA-free Protease Inhibitor | Sigma Aldrich | 04693132001 | |
| Chemical compound, drug | Tetracycline | Sigma Aldrich | 87128 | |
| Chemical compound, drug | HEPES | Sigma Aldrich | H3375 | |
| Chemical compound, drug | Sodium dodecyl sulfate | Sigma Aldrich | 75746 | |
| Chemical compound, drug | PonceauS | Sigma Aldrich | P7170 | |
| Chemical compound, drug | Sodium deoxycholate | Sigma Aldrich | 30970 | |
| Chemical compound, drug | Naphtol blue black | Sigma Aldrich | N3393 | |
| Chemical compound, drug | Adenosine triphosphate | Sigma Aldrich | A2383 | |
| Chemical compound, drug | MG132 | Sigma Aldrich | M7449, 474787 | |
| Chemical compound, drug | NP-40 | Sigma Aldrich | I8896 | |
| Chemical compound, drug | KCl | Sigma Aldrich | I1149 | |
| Chemical compound, drug | Iodoacetamide | Sigma Aldrich | I8896 | |
| Chemical compound, drug | Dimethyl sulfoxide | Sigma Aldrich | D2438 | |
| Chemical compound, drug | Cycloheximide | Sigma Aldrich | C7698 | |
| Chemical compound, drug | Hygromycin B | Thermo Fisher Scientific | 10687010 | |
| Chemical compound, drug | Zeocin | Thermo Fisher Scientific | R25001 | |
| Chemical compound, drug | Sulfo-NHS-Acetate | Thermo Fisher Scientific | 20217 | |
| Chemical compound, drug | DAPI (4',6-Diamidino-2-Phenylindole, Dihydrochloride) | Thermo Fisher Scientific | D1306 | |
| Chemical compound, drug | Permafluor mounting medium | Thermo Fisher Scientific | TA-006-FM | |
| Chemical compound, drug | Poly-D-Lysine | Thermo Fisher Scientific | A3890401 | |
| Chemical compound, drug | Blasticidin | Thermo Fisher Scientific | R21001 | |

*Appendix 1 Continued on next page*

*Appendix 1 Continued*

| Reagent type (species) or resource | Designation | Source or reference | Identifiers | Additional information |
|---|---|---|---|---|
| Chemical compound, drug | EDTA | Carl Roth | 8043.2 | |
| Chemical compound, drug | EGTA | Carl Roth | 3054.1 | |
| Chemical compound, drug | NaCl | Carl Roth | 3957.1 | |
| Chemical compound, drug | Triton X-100 | Carl Roth | 3051.3 | |
| Chemical compound, drug | Tris | Carl Roth | 4855.2 | |
| Chemical compound, drug | Glycerin | Carl Roth | 7533.1 | |
| Chemical compound, drug | β-mercaptoethanol | Carl Roth | 4227.3 | |
| Chemical compound, drug | Tween-20 | Carl Roth | 9127.1 | |
| Chemical compound, drug | Acetic acid | Carl Roth | 6755.1 | |
| Chemical compound, drug | Ammonium bicarbonate | Carl Roth | T871.2 | |
| Chemical compound, drug | Formic acid | Carl Roth | 4724.3 | |
| Chemical compound, drug | Formaldehyde | Carl Roth | CP10.1 | |
| Chemical compound, drug | Methanol | Biosolve | 0013684102BS | |
| Chemical compound, drug | Acetone | Biosolve | 0001037801BS | |
| Chemical compound, drug | Formic acid | Biosolve | 0006914143B5 | |
| Chemical compound, drug | Trifluoroacetic acid | Biosolve | 0020234131BS | |
| Chemical compound, drug | Acetonitrile | Biosolve | 0001204102BS | |
| Chemical compound, drug | 2-propanol | Biosolve | 0016264101BS | |
| Chemical compound, drug | X-tremeGENE 9 DNA Transfection Reagent | Roche | 06365779001 | |
| Chemical compound, drug | Phosphatase inhibitors | Roche | 04906837001 | |
| Chemical compound, drug | protease inhibitors | Roche | 04693159001 | |
| Chemical compound, drug | $MgCl_2$ | Merck | 8.14733.0100 | |
| Chemical compound, drug | Glycine | VWR | 1042011000 | |
| Chemical compound, drug | Urea | Bio Rad | 161–0730 | |

*Appendix 1 Continued on next page*

*Appendix 1 Continued*

| Reagent type (species) or resource | Designation | Source or reference | Identifiers | Additional information |
|---|---|---|---|---|
| Chemical compound, drug | standard chow | ssniff | V1524-786 | |
| Chemical compound, drug | chow with doxycycline | ssniff | A153D00624 | |
| Chemical compound, drug | xylene | VWR | 28973.363 | |
| Chemical compound, drug | ethanol | VWR | 85830.360 | |
| Software, algorithm | Spectronaut | Biognosys | | |
| Other | Dulbecco's modified Eagle's medium (DMEM) high glucose 4.5 g/l | Sigma Aldrich | D6429 | Cell culture media |
| Other | PBS | Sigma Aldrich | D8537 | Buffer |
| Other | Fetal bovine serum | Thermo Fisher Scientific | 10270–106 | Supplement for cell culture media |
| Other | Goat serum | Thermo Fisher Scientific | 31872 | Blocking reagent for immunofluorescence |

