## [Editor Report · eLife assessment]

This study presents an **important** method and resource in cell lines and in mice for mass spectrometry-based identification of interactors of the proteasome, a multi-protein complex with a central role in protein turnover in almost all tissues and cell types. The method presented, including the experimental workflow and analysis pipeline, as well as the several lines of validation provided throughout, is **convincing**. Given the growing interest in protein aggregation and targeted protein degradation modalities, this work will be of interest to a broad spectrum of basic cell biologists and translational researchers.

---

## [Referee Report · Reviewer #2 (Public Review)]

Summary

In this work, Bartolome and colleagues develop a new approach to identify proteasome interacting proteins and substrates. The approach is based on fusing proteasome subunits with a biotin ligase that will label proteins that come in close physical distance of the ligase. These biotin-labeled proteins (or their resulting tryptic peptides) can be affinity purified using streptavidin and identified by mass spectrometry.

This elegant solution was able to identify a large proportion of known proteasome interactors, as well as multiple potential new interactors. Combining this approach with a proteasome inhibitor allowed also for the enrichment of substrates, due to increased contact time between substrates and the proteasome. Again, the authors were able to identify novel substrates. Finally, the authors implemented this strategy in vivo, providing the hints for potential tissue-specific proteasome interactors.

This novel strategy provides an additional approach to identify new proteasome substrates, which can be particularly powerful for low abundant proteins, e.g., transcription factors. The possibility to implement it in vivo in specific cell types opens the possibility for identifying proteasome interactors in small cell subpopulations or in subpopulations involved in disease.

Strengths

The authors carefully characterized their genetically engineered proteasome-biotin ligase fusions to ensure that proteasome structure and activity was not altered. This is key to ensure that the proteins identified to interact with the proteasome reflect interactions that occur under physiological conditions.

The authors implemented an algorithm that controls the false positive rate of the identified interactors of the proteasome. This is an important aspect to avoid spending time on the characterization of potential interactors that are just an artifact of the experimental setup.

The addition of a proteasome inhibitor allowed the authors to identify substrates of the proteasome. Although there are other strategies to do this (e.g., affinity purification of Gly-Gly modified peptides, which is a marker for ubiquitination), this additional approach can highlight currently unknown substrates. One example are low abundance proteins, such as transcription factors.

The overall strategy developed by the authors can be implemented in vivo, which opens for the possibility of determining cell type-specific proteasome interactors (and perhaps substrates).

Weaknesses

There is a proportion (approximately 38%) of the PSMA4-biotin ligase fusion that remains unassembled (i.e., not part of the functional proteasome) and that can contribute to a small proportion of false positive interactions.

---

## [Referee Report · Reviewer #3 (Public Review)]

Summary:

Bartolome et al. present ProteasomeID, a novel method to identify components, interactors, and (potentially) substrates of the proteasome in cell lines and mouse models. As a major protein degradation machine that is highly conserved across eukaryotes, the proteasome has historically been assumed to be relatively homogeneous across biological scales (with few notable exceptions, e.g., immunoproteasomes and thymoproteasomes). However, a growing body of evidence suggests that there is some degree of heterogeneity in the composition of proteasomes across cell tissues, and can be highly dynamic in response to physiologic and pathologic stimuli. This work provides a methodological framework for investigating such sources of variation. The authors start by adapting the increasingly popular biotin ligation strategy for labelling proteins coming into close proximity with one of three different subunits of the proteasome, before proceeding with PSMA4 for further development and analysis based on their preliminary labelling data. In a series of well-constructed and convincing validation experiments, the authors go on to show that the tagged PSMA4 construct can be incorporated into functional proteasomes, and is able to label a broad set of known proteasome components and interacting proteins in HEK293T cells. They also attempt to identify novel proteasomal degradation substrates with ProteasomeID; while this was convincing for known substrates with particularly short half-lives, the results for substrates with longer half-lives were less clear. One of the most compelling results was from a similar experiment to confirm proteasomal degradation induced by a BRD-targeting PROTAC, which I think is likely to be of keen interest to the targeted degradation community. Finally, the authors establish a ProteasomeID mouse model, and demonstrate its utility across several tissues.

Strengths:

(1) ProteasomeID itself is an important step forward for researchers with an interest in protein turnover across biological scales (e.g., in sub-cellular compartments, in cells, in tissues, and whole organisms). I especially see interest from two communities: those studying fundamental proteostasis in physiological and pathologic processes (e.g., ageing; tissue-specific protein aggregation diseases), and those developing targeted protein degradation modalities (e.g., PROTACs; molecular glues). All the datasets generated and deposited here are likely to provide a rich resource to both. The HEK293T cell line data are a valuable proof-of-concept to allow expansion into more biologically-relevant cell culture settings; however, I envision the greatest innovation here to be the mouse model. For example, in the targeted protein degradation space, two major hurdles in early-stage pre-clinical development are (i) evaluation of degradation efficacy across disease-relevant tissues, and (ii) toxicity and safety implications caused by off-target degradation, e.g., of newly-identified molecular glues and/or in particularly-sensitive tissues. The ProteasomeID mouse allows early in vivo assessment of both these questions. The results of the BRD PROTAC experiment in 293T cells provides an excellent in vitro proof-of-concept for this approach.

(2) The mass-spectrometry-based proteomics workflows used and presented throughout the manuscript are robust, rigorous, and convincing. For example, the algorithm the authors use for defining enrichment score cut-offs are logical and based on rational models, rather than on arbitrary cut-offs that are common for similar proteomics studies. The construction (and subsequent validation) of both BirA*- and miniTurbo- tagged PSMA4 variants also increases the utility of the method, allowing researchers to choose the variant with the labelling time-scale required for their particular research question.

(3) The optimised BioID and TurboID protocol the authors develop (summarised in Fig. S2A) and validate (Fig. S2B-D) is likely to be of broad interest to cell and molecular biologists beyond the protein degradation field, given that proximity labelling is a current gold-standard in global protein:protein interaction profiling.

Limitations:

I think the authors do an excellent job in highlighting the limitations of ProteasomeID throughout the Results and Discussion. I do have some specific comments that might provide additional context for the reader.

(1) The authors do a good job in showing that a substantial proportion of PSMA4-BirA* is incorporated into functional proteasome particles; however, it is not immediately clear to me how much background (false-positive IDs) might be contributed by the ~40 % of PSMA4-BirA* that is not incorporated into the mature core particle (based on the BirA* SEC-MS traces in Fig. 2b and S3b, i.e., the large peak ~ fraction 20). Are there any bands lower down in the native gel shown in Fig. 2c, i.e., corresponding to lower molecular weight complexes or monomeric PSMA4-BirA*? The enrichment of proteasome assembly factors in all the ProteasomeID experiments might suggest the presence of assembly intermediates, which might themselves become substrates for proteasomal degradation (as has been shown for other incompletely-assembled protein complexes, e.g., the ribosome, TRiC/CCT).

(2) Although the authors attempt to show that BirA* tagging of PSMA4 does not interfere with proteasome activity (Fig. 2e-f), I think the experimental evidence for this is incomplete. They show that the overall chymotrypsin-like activity (attributable to PSMB5) in cells expressing PSMA4-BirA* is not markedly reduced compared with control BirA*-expressing cells. However, they do not show that the activity of the specific proteasome sub-population that contains PSMA4-BirA* is unaffected (e.g., by purifying this sub-population via the Flag tag). The proteasome activity of the sub-population of wild-type proteasome complexes that do not contain the PSMA4-BirA* (~50%, based on the earlier immunoblots) could account for the entire chymotrypsin-like activity-especially in the context of HEK293T cells, where steady-state proteasome levels are unlikely to be limiting. It would also be useful to assess any changes in tryspin- and caspase- like activities, especially as tagging of PSMA4 could conceivably interfere with the activity of some PSMB subunits, but not others.

(3) I was left slightly unsure as to the general utility of ProteasomeID for identifying novel proteasomal substrates in homeostatic conditions--especially for proteins with longer half-lives. The cycloheximide chases in Fig. 4g/S4j are clear for MYC and TIGD5 (which have short half-lives), but are not so clear for ARMC6 and BRAT1: the reduction in the bands are modest, and might have been clearer with longer "chase" time-points. Furthermore, classifying candidates based on enrichment following proteasome inhibition with MG-132 have the potential to lead to a high number of false positives. ProteasomeID's utility in identifying potential substrates in more targeted settings (e.g., molecular glues, off-target PROTAC substrates) is far more apparent.

---

## [Author Response]

The following is the authors’ response to the original reviews.

**Reviewer #1 (Recommendations For The Authors):**
Specific comments to improve the quality of the work:(1) The choice of subunits to tag are really not ideal. In the available structures of the human proteasome, The C-terminus of Rpn3/PSMD3 points directly toward the ATPase pore and is likely to disrupt the structure and/or dynamics of the proteasome during proteolysis (see comments regarding controls for functionality below). Similarly, the C-terminal tail of Rpt1/PSMC2 has a key role in the opening of the 20S core particle gate for substrate translocation and processing (see 2018 Nature Communications, 9:1360 and 2018 Cell Reports 24:1301-1315), and Alpha3/PSMA4 can be substituted by a second copy of Alpha4/PSMA7 in some conditions (although tagging Alpha3/PSMA4 would admittedly provide a picture of the canonical proteasome interactome while actively excluding the interactome of the non-canonical proteasomes that form via replacement of Alpha3/PSMA4). Comparison of these cell lines with lines harboring tags on subunits that are commonly used for tagging in the field because of a lack of impacts, such as the N-terminus of Rpn1/PSMD2, the C-terminus of Rpn11/PSMD14, and the C-terminus of Beta4/PSMB2 would help instill confidence that the interactome reported largely arises from mature, functional proteasomes rather than subcomplexes, defective proteasomes, or other species that may occur due to tagging at these positions.

We thank the reviewer for pointing this out. The original purpose of our strategy was to establish proximity labeling of proteasomes to enable applications both in cell culture and in vivo. The choice of PSMA4 and PSMC2 was dictated by previous successful tagging with GFP in mammalian cells (Salomons et al., Exp Cell Res 2010)(Bingol and Schuman, Nature 2006). However, the choice of C-terminal PSMC2 might have been not optimal. HEK293 cells overexpressing PSMC2-BirA* show slower growth and the BioID data retrieve higher enrichment of assembly factors suggesting slower assembly of this fusion protein in proteasome. Although we did not observe a negative impact on overall proteasome activity and PSMC2-BirA* was (at least in part) incorporated into fully assembled proteasomes as indicated by enrichment of 20S proteins.We apologize for not making it clear that we labeled the N-terminus of PSMD3/Rpn3 and not the C-terminus (Figure 1a and S1a). Therefore, we included in Figure S1a of the revised manuscript structures of the proteasome where the tagged subunit termini are highlighted: C-terminus for PSMA4 and PSMC2 and N-terminus for PSMD3.Additionally, we would like to point out that, differently from PSMC2-BirA*, cells expressing BirA*-PSMD3 did not show slower growth, and BioID data showed a more homogenous enrichment of both 19S and 20S proteins, as compared to PSMC2-BirA* (Figure 1D and 1E). However, the overall level of enrichment of proteasome subunits was not comparable to PSMA4-BirA* and, therefore, we opted for focusing the rest of the manuscript on this construct.

In support of this point, the data provided in Figure 1E in which the change in the abundances of each proteasome subunit in the tagged line vs. the BirA control line demonstrates substantial enrichment of the subcomplexes of the proteasome that are tagged in each case; this effect may represent the known feedback-mediated upregulation of new proteasome subunit synthesis that occurs when proteasomal proteolysis is impaired, or alternatively, the accumulation of subcomplexes containing the tagged subunit that cannot readily incorporate into mature proteasomes. Acknowledging this limitation in the text would be valuable to readers who are less familiar with the proteasome.

We would like to clarify that the data shown in Figure 1E do not represent whole proteome data, but rather log2 fold changes vs. BirA* control calculated on streptavidin enrichment samples. The differences in the enrichment of the various subcomplexes between cell lines derives from the fact that the effect size of the enrichment depends on both protein abundance in the isolated complexes, but also on the efficiency of biotinylation. The latter will be higher for proteins located in closer proximity to the bait. A similar observation was pointed out in a recent publication (PMID: 36410438) that compared BioID and Co-IP for the same bait. When a component of the nuclear pore complex (Nup158) was analyzed by BioID only the more proximal proteins were enriched as compared to the whole complex in Co-IP data (Author response image 1):

**Author response image 1. sa3fig1:** Proteins identified in the NUP158 BioID or pulldown experiments are filled in red or light red for significance intervals A or B, respectively. The bait protein NUP158 is filled in yellow. Proteins enriched in the pulldown falling outside the SigA/B cutoff are filled in gray. NPC, nuclear pore complex. SigA, significant class A; SigB, significant class B. Reproduced from Figure 6 of Moreira CMDN et al., 2023 (PMID: 36410438).

However, we would like to point out that despite quantitative differences between different proteasome subunits, both 19S and 20S proteins were found to be strongly enriched (typically >2 fold) in all the constructs compared to BirA* control line (Figure 1E). This indicates that at least a fraction of all the tagged subunits are incorporated into fully assembled proteasomes.

Regarding the upregulation of proteasome subunits as a consequence of proteasome dysfunction, we did not find evidence of this, at least in the case of PSMA4. The immunoblot shown in Figure 2A and its quantification in S3A indicate no increased abundance of endogenous PSMA4 upon tetracycline induction of PSMA4-BirA*.

(2) The use of myc as a substrate of the proteasome for demonstration that proteolysis is unaffected is perhaps not ideal. Myc is known to be degraded via both ubiquitin-dependent and ubiquitin-independent mechanisms, such that disruption of one means of degradation (e.g., ubiquitin-dependent degradation) via a given tag could potentially be compensated by another. A good example of this is that the C-terminal tagging of PSMC2/Rpt1 is likely to disrupt interaction between the core particle and the regulatory particle (as suggested in Fig. 1D); this may free up the core particle for ubiquitin-independent degradation of myc.Aside from using specific reporters for ubiquitin-dependent vs. independent degradation or a larger panel of known substrates, analysis of the abundance of K48-ubiquitinated proteins in the control vs. tag lines would provide additional evidence as to whether or not proteolysis is generally perturbed in the tag lines.

We thank the reviewer for this suggestion. We have included an immunoblot analysis showing that the levels of K48 ubiquitylation (Figure S3d) are not affected by the expression of tagged PSMA4.

(3) On pg. 8 near the bottom, the authors accidentally refer to ARMC6 as ARMC1 in one instance.

We have corrected the mistake.

(4) On pg. 10, the authors explain that they analyzed the interactome for all major mouse organs except the brain; although they explain in the discussion section why the brain was excluded, including this explanation on pg. 10 here instead of in the discussion might be a better place to discuss this.

We moved the explanation from the discussion to the results part.

**Reviewer #2 (Recommendations For The Authors):**
(1) Perhaps the authors can quantify the fraction of unassembled PSMA4-BirA* from the SEC experiment (Fig. 2b) to give the readers a feeling for how large a problem this could be.

The percentages based on Area Under the Curve calculations have been added to Figure S3b.

(2) Do the authors observe any difference in the enrichment scores between proteins that are known to interact with the proteasome vs proteins that the authors can justify as "interactors of interactors" vs the completely new potential interactors? This could be an interesting way to show that the potential new interactors are not simply because of poor false positive rate calibration, but that they behave in the same way as the other populations.

We thank the reviewer for this suggestion. We analyzed the enrichment scores for 20S proteasome subunits, known PIPs, first neighbors and the remaining enriched proteins. The remaining proteins (potential new interactors) have very similar scores as the first neighbors of known interactors. This plot has been added to Figure S3g.

(3) Did the authors try to train a logistic model for the miniTurbo experiments, like it was done for the BirA* experiments? Perhaps combining the results of both experiments would yield higher confidence on the proteasome interactors.

Following the reviewers suggestion, we applied the classifier on the dataset of the comparison between miniTurbo and PSMA-miniTurbo. We found a clear separation between the FPR and the TPR with 136 protein groups enriched in PSMA-miniTurbo. We have added the classifier and corresponding ROC curve to Figure S4f and S4g.

75 protein groups were found to be enriched for both PSMA4-BirA* and PSMA4-miniTurbo (Author response image 2), including the proteasome core particles, regulatory particles, known interactors and potential new interactors. As we focused more on the identification of substrates with PSMA4-miniTurbo, we did not pursue these overlapping protein groups further, but rather used the comparison to the mouse model to identify potential new interactors.

**Author response image 2. sa3fig2:** Overlap between ProteasomeID enriched proteins (fpr<0.05) between PSMA4-BirA* and PSMA4-miniTurbo.

(4) Perhaps this is already known, but did the authors check if MG132 affect proteasome assembly? The authors could for example repeat their SEC experiments in the presence of MG132.

We thank the reviewer for the suggestion, however to our knowledge there are no reports that MG132 has an effect on the assembly of the proteasome. MG132 is one of the most used proteasome inhibitors in basic research and as such has been extensively characterized in the last 3 decades. The small peptide aldehyde acts as a substrate analogue and binds directly to the active site of the protease PSMB5/β5. We therefore think it is unlikely that MG132 is interfering with the assembly of the proteasome.

(5) Minor comment: at the bottom of page 8, the authors probably mean ARMC6 and not ARMC1.

We have corrected the mistake.

(6) It would be interesting to expand the analysis of the already acquired in vivo data to try to identify tissue-specific proteasome interactors. Can the authors draw a four-way Venn diagram with the interactors of each tissue?

We thank the reviewer for this suggestion. We have generated an UpSet plot showing the overlap of ProteasomeID enriched proteins in the four tissues that gave us meaningful results (Author response image 3). In order to investigate whether the observed differences in ProteasomeID enriched proteins could be meaningful in terms of proteasome biology, we have highlighted proteins belonging to the UPS that show tissue specific enrichments. We found proteasome activators such as PSME1/PA28alpha and PSME2/PA28beta to enrich preferentially in kidney and liver, respectively, as well as multiple deubiquitinases to enrich preferentially in the heart. These differences might be related to the specific cellular composition of the different tissues, e.g., number of immune cells present, or the tissue-specific interaction of proteasomes with enzymes involved in the ubiquitin cycle. Given the rather preliminary nature of these findings, we have opted for not including this figure in the main manuscript, but rather include it only in this rebuttal letter.

**Author response image 3. sa3fig3:** Upset plot showing overlap between ProteasomeID enriched proteins in different mouse organs.

**Reviewer #3 (Recommendations For The Authors):**
(1) In the first paragraph of the Introduction, the authors link cellular senescence caused by partial proteasome inhibition with the efficacy of proteasome inhibitors in cancer therapy. Although this is an interesting hypothesis, I am not aware of any direct evidence for this; rather, I believe the efficacy of bortezomib/carfilzomib in haematological malignancies is most commonly attributed to these cells having adapted to high levels of proteotoxic stress (e.g., chronic unfolded protein response activation). I would suggest rephrasing this sentence.

We thank the reviewer for the comment and have amended the introduction.

(2) For the initial validation experiments (e.g., Fig. 1B), have the authors checked what level of Streptavidin signal is obtained with "+ bio, - tet" ? Although I accept that the induction of PSMA4-BirA* upon doxycycline addition is clear from the anti-Flag blots, it would still be informative to ascertain what level of background labelling is obtained without induction (but in the presence of exogenous biotin).

We tested four different conditions +/- tet and +/- biotin (24h) in PSMA4-BirA* cell lines (Author response image 4). As expected, biotinylation was most pronounced when tet and biotin were added. When biotin was omitted, streptavidin signal was the lowest regardless of the addition of tet. Compared to the -biotin conditions, a slight increase of streptavidin signal could be observed when biotin was added but tet was not added. This could be either due to the promoter leaking (PMID: 12869186) or traces of tetracycline in the FBS we used, as we did not specifically use tet-free FBS for our experiments.

**Author response image 4. sa3fig4:** Streptavidin-HRP immunoblot following induction of BirA* fusion proteins with tetracycline (+tet) and supplementation of biotin (+bio). For the sample used as expression control tetracycline was omitted (-tet). To test background biotinylation, biotin supplementation was omitted (-bio). Immunoblot against BirA* and PSMA was used to verify induction of fusion proteins, while GAPDH was used as loading control.

(3) For the proteasome structure models in Fig. 1D, a scale bar would be useful to inform the reader of the expected 10 nm labelling radius (as the authors have done later, in Fig. 2D).

We have added 10 nm scale bars to Figure 1d.

(4) In the "Identification of proteasome substrates by ProteasomeID" Results subsection, I believe there is a typo where the authors refer to ARMC1 instead of ARMC6.

We have corrected the mistake.

(5) I think Fig. S5 was one of the most compelling in the manuscript. Given the interest in confirming on-target efficacy of targeted degradation modalities, as well as identifying potential off-target effects early-on in development, I would consider promoting this out of the supplement.

We thank the reviewer for the comment and share the excitement about using ProteasomeID for targeted degradation screening. We have moved the data on PROTACs (Figure S5) into a new main Figure 5.

In addition, in relation to the comment of this reviewer regarding the detection of endogenous substrates, we have now included validation for one more hit emerging from our analysis (TIGD5) and included the results in Figure 4f, 4g and S4j.